# The mitochondrial Hsp70 controls the assembly of the $F_1F_O$-ATP synthase

Jiyao Song [1,2], Liesa Steidle[2], Isabelle Steymans[1], Jasjot Singh [1], Anne Sanner [1], Lena Böttinger[2], Dominic Winter [1] & Thomas Becker [1] ✉

The mitochondrial $F_1F_O$-ATP synthase produces the bulk of cellular ATP. The soluble $F_1$ domain contains the catalytic head that is linked via the central stalk and the peripheral stalk to the membrane embedded rotor of the $F_O$ domain. The assembly of the $F_1$ domain and its linkage to the peripheral stalk is poorly understood. Here we show a dual function of the mitochondrial Hsp70 (mtHsp70) in the formation of the ATP synthase. First, it cooperates with the assembly factors Atp11 and Atp12 to form the $F_1$ domain of the ATP synthase. Second, the chaperone transfers Atp5 into the assembly line to link the catalytic head with the peripheral stalk. Inactivation of mtHsp70 leads to integration of assembly-defective Atp5 variants into the mature complex, reflecting a quality control function of the chaperone. Thus, mtHsp70 acts as an assembly and quality control factor in the biogenesis of the $F_1F_O$-ATP synthase.

Mitochondria contain about 900 and 1200 proteins in baker´s yeast *Saccharomyces cerevisiae* and humans, respectively[1–4]. About 99% of the mitochondrial proteins are synthesised as precursors on cytosolic ribosomes. The presequence pathway represents the major protein import route into mitochondria. Precursors contain a cleavable presequence and are transported across the outer membrane via the translocase of the outer membrane (TOM complex). Subsequently, the presequence translocase (TIM23 complex) imports these proteins into the inner membrane and matrix[5–8]. The membrane potential drives protein translocation via the TIM23 complex[9,10]. Transport into the mitochondrial matrix additionally requires the ATP-consuming activity of the presequence translocase-associated motor (PAM)[11–13]. The mitochondrial Hsp70 (mtHsp70) is the core subunit of the PAM module and drives protein translocation by a cycle of substrate binding and release. The chaperone consists of an N-terminal ATPase domain and C-terminal substrate binding domain. ATP-binding and hydrolysis modulate the closing of the substrate binding domain. In the ADP-bound stage, mtHsp70 binds to client proteins with a high affinity, while the exchange of ADP with ATP releases the trapped substrate[14,15]. Co-chaperones control the reaction cycle of mtHsp70. Within the PAM module, the J-protein Pam18 (also termed Tim14) forms a heterodimer with Pam16 (also termed Tim16) and stimulates the ATPase activity of mtHsp70, while Mge1 induces the ADP to ATP

exchange[16–23]. MtHsp70 performs further critical functions to maintain mitochondrial proteostasis. The chaperone folds imported proteins, facilitates the degradation of misfolded proteins and promotes biogenesis of selected protein complex subunits[11,13,24–34].

The mitochondrial $F_1F_O$-ATP synthase (ATP synthase) utilises a proton gradient across the inner membrane that is generated by activity of the respiratory chain to synthesise the main fraction of cellular ATP. The ATP synthase consists of a membrane-integrated $F_O$ domain, the peripheral stalk and the soluble $F_1$ domain composed of the catalytic head and the central stalk[35–39]. In yeast, the three mitochondrially encoded subunits form the core of the membrane-bound $F_O$ domain (Atp6 (subunit a), Atp8 (subunit 8) and Atp9 (subunit c)), which catalyses the proton transport across the inner membrane. Atp6 allows proton transport that is coupled to rotation of the rotor ring that is composed of ten Atp9 subunits[40]. The movement of the rotor is transmitted via Atp3 (subunit γ) of the central stalk to the catalytic head[41]. Three heterodimers of Atp1 (subunit α) and Atp2 (subunit β) form the three catalytic centres that surround Atp3. The rotation of Atp3 induces conformational changes in three reaction centres, leading to ATP production. The peripheral stalk binds the catalytic head to prevent its rotation. The membrane-anchored Atp4 (subunit b) together with the soluble proteins Atp14 (subunit h), Atp7 (subunit d) and Atp5 (oligomycin sensitivity conferring protein

[1]Institute for Biochemistry and Molecular Biology, Faculty of Medicine, University of Bonn, Bonn, Germany. [2]Institute for Biochemistry and Molecular Biology, Faculty of Medicine, University of Freiburg, Freiburg, Germany. ✉e-mail: thbecker@uni-bonn.de

(OSCP)) form the peripheral stalk. Atp5 links the peripheral stalk to the catalytic head. Finally, membrane-bound supernumerary sub-units mediate dimerisation of the ATP synthase, which promotes the formation of the cristae[35,42,43].

The formation of the ATP synthase is a highly complicated process and involves the coordinated assembly of three mitochondrially encoded subunits with fourteen nuclear-encoded components[44–47]. According to current models, the assembly process occurs in a step-wise manner and involves the intermediate formation of assembly modules[48]. Our understanding about the molecular mechanisms of the formation of the $F_1$-domain and its subsequent association with the peripheral stalk is limited. Only a few assembly factors have been reported. In the matrix, Atp11 and Atp12 bind to unassembled Atp2 and Atp1, respectively, to prevent their aggregation[49–51]. The matrix protein Fmc1 (formation of mitochondrial complexes 1) is required to maintain normal Atp12 levels[52]. Consequently, Atp1 and Atp2 aggregate in the absence of Fmc1 at higher temperature[52]. How Atp1 and Atp2 assemble into the $F_1$ domain remains unknown. The inner membrane assembly complex (INAC) promotes the formation of the peripheral stalk and facilitates its association with the membrane-bound rotor module[53,54]. INAC consists of the two single-spanning inner membrane proteins Ina17 and Ina22 that form a protein complex of around 300 kDa on blue native gels[53,54]. However, the ATP synthase is formed in the absence of INAC, indicating that further assembly factors remain to be identified. Particularly, the assembly of Atp5 remains unknown, although the addition of Atp5 is critical to link the $F_1$ domain with the peripheral stalk.

In this work, we report an unexpected dual role of mtHsp70 in the formation of the ATP synthase. The chaperone cooperates with Atp11 and Atp12 to build up the $F_1$ domain. Furthermore, mtHsp70 transfers Atp5 to the INA complex to allow linkage of the $F_1$-domain and the peripheral stalk. Remarkably, mutations of mtHsp70 that impair sub-strate binding allow the integration of an assembly-defective Atp5 mutant into the ATP synthase. We conclude that mtHsp70 functions as assembly and quality control factor in the formation of the ATP synthase.

## Results

### mtHsp70 binds to several subunits of the ATP synthase

We aimed to study proteomic changes upon deletion of mitochondrially encoded OXPHOS components in $\rho^0$ cells that have lost the mitochondrial DNA. To generate $\rho^0$ cells, wild-type cells were treated with ethidium bromide[55] and the lack of mitochondria DNA was confirmed by PCR (data not shown). The cells were differentially labelled by stable isotope labelling in cell culture (SILAC) with either heavy isotopes of arginine and lysine ($\rho^0$) or with corresponding light amino acids (wild-type). Subsequently, mitochondria were isolated and analysed by mass spectrometry. We found that the majority of nuclear-encoded subunits of the respiratory chain and the ATP synthase were strongly reduced in $\rho^0$ mitochondria (Fig. 1a, Supplementary Fig. 1a and Supplementary Data 1)[56]. Western blot analysis confirmed that mitochondrially encoded subunits of complex IV were absent, but the steady-state levels of nuclear-encoded subunits of complex IV and of the ATP synthase were affected differentially. For the ATP synthase, we found that Atp1 and Atp2 were only mildly reduced, while the levels of Atp5 and Atp4 were strongly decreased (Supplementary Fig. 1b). For comparison, the steady-state levels of Tom40 and mitochondrial (mt) Hsp70 were unchanged (Supplementary Fig. 1b). We used blue native electrophoresis to study mitochondrial protein complexes in $\rho^0$ mitochondria. As expected, the respiratory chain supercomplexes of complexes III and IV[28,57] were absent $\rho^0$ mitochondria, while the TOM complex was present in comparable amounts to wild-type mitochondria (Supplementary Fig. 1c). A monomer and dimer of the ATP synthase as well as the $F_1$ domain can be analysed by blue native electrophoresis[53,54,57]. Monomer and dimer of the ATP synthase were

absent in the mutant mitochondria, while the $F_1$ domain of the ATP synthase was still formed and present in the soluble fraction (Supplementary Fig. 1c, d). The ATPase activity of the $F_1$ domain is important to maintain the membrane potential in $\rho^0$ mitochondria[47,58,59], explaining why the $F_1$ domain is assembled in the absence of mitochondrially encoded subunits.

We wondered whether unassembled ATP synthase subunits accumulate at mtHsp70 in $\rho^0$ mitochondria. We expressed His-tagged mtHsp70 in wild-type ($\rho^+$) and $\rho^0$ cells and labelled the cells with either heavy isotopes (mtHsp70$_{His}$ $\rho^0$) or medium isotopes (mtHsp70$_{His}$ $\rho^+$) of arginine and lysine. Subsequently, mtHsp70-bound proteins were purified from isolated mitochondria utilising the His-tag of the chaperone. The two elution fractions from mtHsp70$_{His}$ $\rho^+$ and mtHsp70$_{His}$ $\rho^0$ mitochondria were mixed and analysed by mass spectrometry. We determined the ratios of the proteins bound to mtHsp70$_{His}$ in $\rho^0$ and $\rho^+$ mitochondria (fold change) (Fig. 1b, Supplementary Data 2). Following this strategy, we detected strong enrichment of subunits of the $F_1$ domain (Atp1, Atp2, Atp3 and Atp15) and the peripheral stalk (Atp4, Atp5, Atp7) of the ATP synthase in the elution fraction compared to wild-type mitochondria. We also found an increased binding of the assembly factors Atp11 and Atp12 to mtHsp70$_{His}$ (Fig. 1b). To exclude secondary effects due to the expression of a His-tagged mtHsp70 in $\rho^0$ cells, we also analysed mtHsp70-bound proteins in wild-type and $\rho^0$ mitochondria after purification via Mge1$_{His}$ that was coupled to Ni-NTA agarose[28,60]. Previously, we have established that such Mge1 affinity matrix can be used to purify partner proteins of mtHsp70[28,29]. Following this strategy, we detected a strong enrichment of subunits of the ATP synthase and other respiratory chain subunits at mtHsp70 in $\rho^0$ mitochondria, while the binding of TIM23 and PAM subunits remain largely unaffected (Fig. 1c, Supplementary Data 3). We analysed both affinity purifications via Western blotting and could corroborate the strong accumulation of Atp1, Atp2 and Atp5 of the ATP synthase at mtHsp70 in $\rho^0$ mitochondria, whereas Tim17, Tim44, Pam16 and Pam18 were co-purified with similar efficiency like in wild-type mitochondria (Fig. 1d, e). Our affinity purifications via mtHsp70$_{His}$ and Mge1$_{His}$ affinity matrix revealed that a small fraction of subunits of the ATP synthase associates with mtHsp70 also in wild-type mitochondria (Fig. 1d, e, lane 4). Supporting this observation, mtHsp70 was pulled down by His-tagged Atp1 (Fig. 1f, lane 4). We tested whether the presequence translocase-associated pool of mtHsp70 binds to ATP synthase subunits. However, we could not co-purify ATP synthase along with His-tagged Tim17 and vice versa we did not observe any binding of TIM23 complex subunits to Atp1$_{His}$ (Supplementary Fig. 1e, f), indicating that the presequence translocase-associated mtHsp70 is distinct from the mtHsp70 fraction that binds to ATP synthase subunits. We conclude that subunits of the $F_1$ domain and the peripheral stalk of the ATP synthase bind to mtHsp70 and accumulate at the chaperone in the absence of a mature enzyme.

To study whether mtHsp70 binds to Atp1 and Atp2 that are assembled in the $F_1$ domain, we studied this interaction in mitochondria from cells lacking Atp3 of the central stalk. In the absence of Atp3, the $F_1$ domain is lost and Atp1 and Atp2 accumulate in low molecular weight forms on blue native gels (Fig. 2a). Using purification via the Mge1$_{His}$ affinity matrix, we found that Atp1 and Atp2 strongly accumulate at mtHsp70 in $atp3\Delta$ mitochondria (Fig. 2b). In contrast, the chaperone-bound pool of Atp1 and Atp2 is only mildly increased in mutants of the peripheral stalk (Fig. 2c), where a $F_1$ domain can be formed (Fig. 2d). These observations indicate that the chaperone binds predominantly to unassembled Atp1 and Atp2. Interestingly, the mtHsp70-bound Atp5 fraction is strongly enriched in mutants of the central and peripheral stalk (Fig. 2b, c), reflecting that Atp5 requires both domains for its assembly into the ATP synthase. We conclude that unassembled ATP synthase subunits bind to mtHsp70.

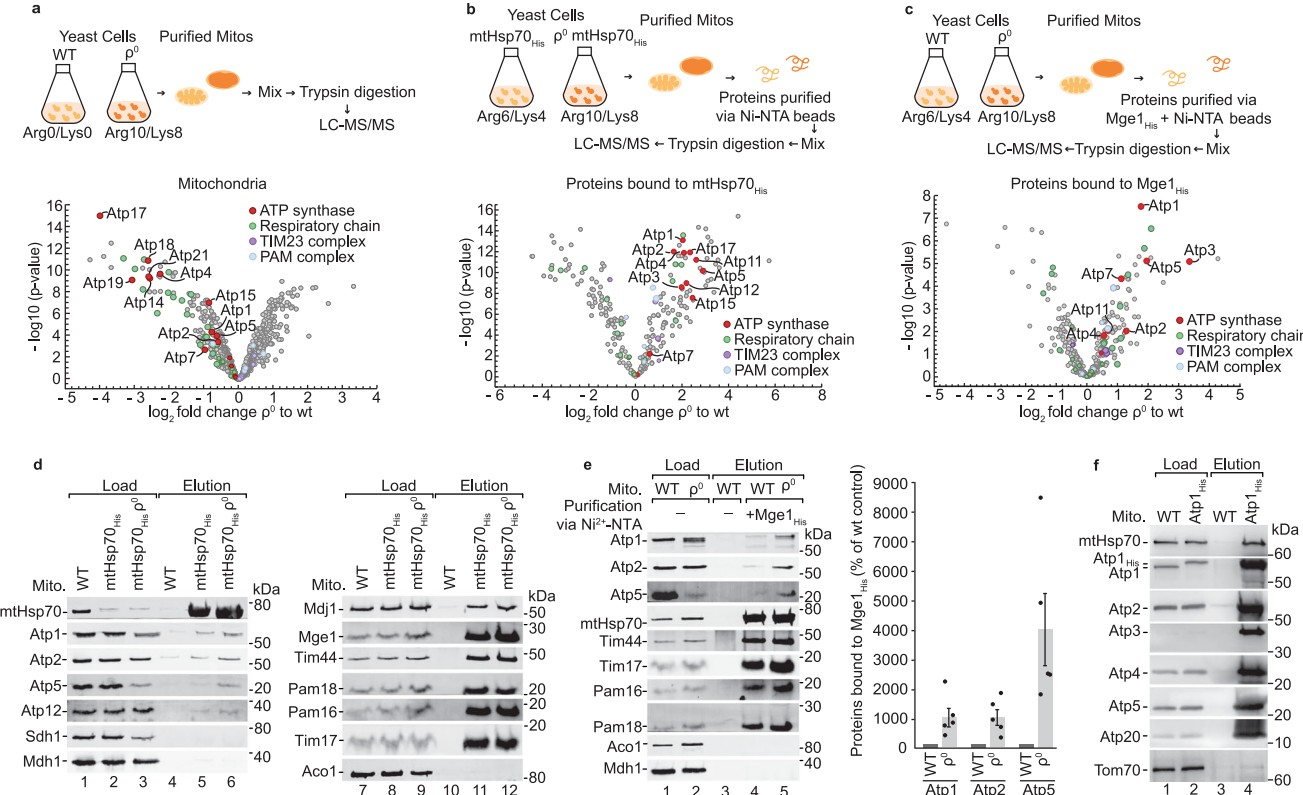

**Fig. 1 | MtHsp70 binds to subunits of the ATP synthase. a** Wild-type (WT) and $\rho^0$ mitochondria were analysed by SILAC labelling and mass spectrometry. Depicted are the mean $\log_2$ value of the fold changes of protein intensities in $\rho^0$ versus WT mitochondria of four replicates, plotted against their statistical significance ($-\log_{10}$ of $p$ values). **b** mtHsp70$_{His}$ and mtHsp70$_{His}$ $\rho^0$ mitochondria from SILAC-labelled cells were subjected to affinity purification via Ni-NTA agarose in the absence of ATP followed by mass spectrometry. Depicted are the mean $\log_2$ values of fold changes of mtHsp70$_{His}$ -bound proteins in $\rho^0$ versus WT background mitochondria of four replicates, plotted against their statistical significance ($-\log_{10}$ of $p$ values). The signals for Lys2 and Ccp1 had unusually high values, indicating an unspecific interaction, and were removed from the plot. **c** Differentially SILAC-labelled WT and $\rho^0$ mitochondrial lysates were subjected to affinity purification via Mge1$_{His}$-coated Ni-NTA agarose in the absence of ATP followed by mass spectrometry. Depicted are the mean $\log_2$ values of fold changes of Mge1$_{His}$-bound proteins in $\rho^0$ versus WT mitochondria of four replicates, plotted against their statistical significance (-$\log_{10}$

of $p$-values). **d** MtHsp70$_{His}$ and mtHsp70$_{His}$ $\rho^0$ mitochondria were subjected to affinity purification via Ni-NTA agarose in the absence of ATP followed by SDS-PAGE and immunodetection. Load: (1% right panel, 2.5% left panel), elution 100%. **e** Left panel, WT and $\rho^0$ mitochondria were subjected to affinity purification via Mge1$_{His}$ in the absence of ATP followed by SDS-PAGE and immunodetection. Load, 1% (Atp1, Atp2, Atp5, mtHsp70, Mdh1) and 2.5% (other proteins), elution 100%. Right panel, Quantification of the co-purified Atp1, Atp2 and Atp5 with Mge1$_{His}$. Depicted are mean values ± SEM of 5 independent experiments. The fractions of Atp1, Atp2 or Atp5 co-eluted with Mge1$_{His}$ in WT mitochondria were set to 100% (control). Subsequently, the amount of Atp1, Atp2 or Atp5 co-eluted with Mge1$_{His}$ in $\rho^0$ was determined and the co-purification efficiency was correlated to WT. Source data are provided as a Source Data file. **f** WT and Atp1$_{His}$ mitochondria were subjected to affinity purification via Ni-NTA agarose followed by immunoblotting. Load: 1%, elution: 100%.

## mtHsp70 promotes the formation of the F$_1$ domain

We wondered whether mtHsp70 promotes the assembly of the F$_1$ domain. MtHsp70 drives import of precursor proteins and their subsequent folding and assembly in the mitochondrial matrix[11,33]. To distinguish between these activities of the chaperone, we used the temperature-sensitive mutant *ssc1-62 (SSC1* is the gene encoding for mtHsp70). When *ssc1-62* mutant cells were grown under permissive conditions, protein import into isolated mitochondria is not impaired[28,29]. However, the formation of cytochrome *c* oxidase was delayed in the mutant mitochondria[28]. Therefore, we isolated mitochondria from cells grown under permissive conditions to investigate whether mtHsp70 plays a role in the assembly of imported Atp1 and Atp2. Under these conditions, the levels of the ATP synthase, TOM complex and Hsp60 complexes were unaffected and mitochondrial proteins were present in comparable amounts as in wild-type mitochondria (Supplementary Fig. 2). For the import assay, [35S]-labelled Atp1 and Atp2 precursors were synthesised in a cell-free translation system in the presence of [35S]methionine and imported into *ssc1-62* mitochondria. The import of Atp1 or Atp2 into isolated mitochondria was monitored by the formation of the processed mature form, which

is blocked upon dissipation of the membrane potential (Fig. 3a). We found that the import of both ATP synthase subunits into in *ssc1-62* mitochondria was mildly accelerated (Fig. 3a, lanes 5-7 and 13-15). We then studied the assembly of Atp1 and Atp2 into the ATP synthase by blue native electrophoresis. The assembly of Atp1 and Atp2 into the monomer and dimer of the ATP synthase occurs with low efficiency in wild-type mitochondria, while the F$_1$ domain is formed efficiently (Fig. 3b, lanes 1–3 and 9-11). In *ssc1-62* mitochondria, the assembly of imported Atp1 and Atp2 into the F$_1$ domain was reduced (Fig. 3b, lanes 5-7 and 13-15). To investigate the role of mtHsp70 for the assembly of the ATP synthase in vivo, we shifted *ssc1-62* cells to non-permissive conditions prior to isolation of mitochondria and analysed the ATP synthase by blue native electrophoresis. We found that the levels of the F$_1$ domain on steady-state levels were strongly reduced. In contrast, the levels of the monomer of the ATP synthase were increased and the dimer mildly decreased (Fig. 3c). We also detected the reduced levels of the F$_1$ domain by using in-gel activity staining of the ATP synthase (Fig. 3d, lanes 4-6)[61,62]. As expected, import and consequently the assembly of Atp1 and Atp2 were strongly impaired in *ssc1-62* mitochondria when the cells were heat-stressed before isolation of the

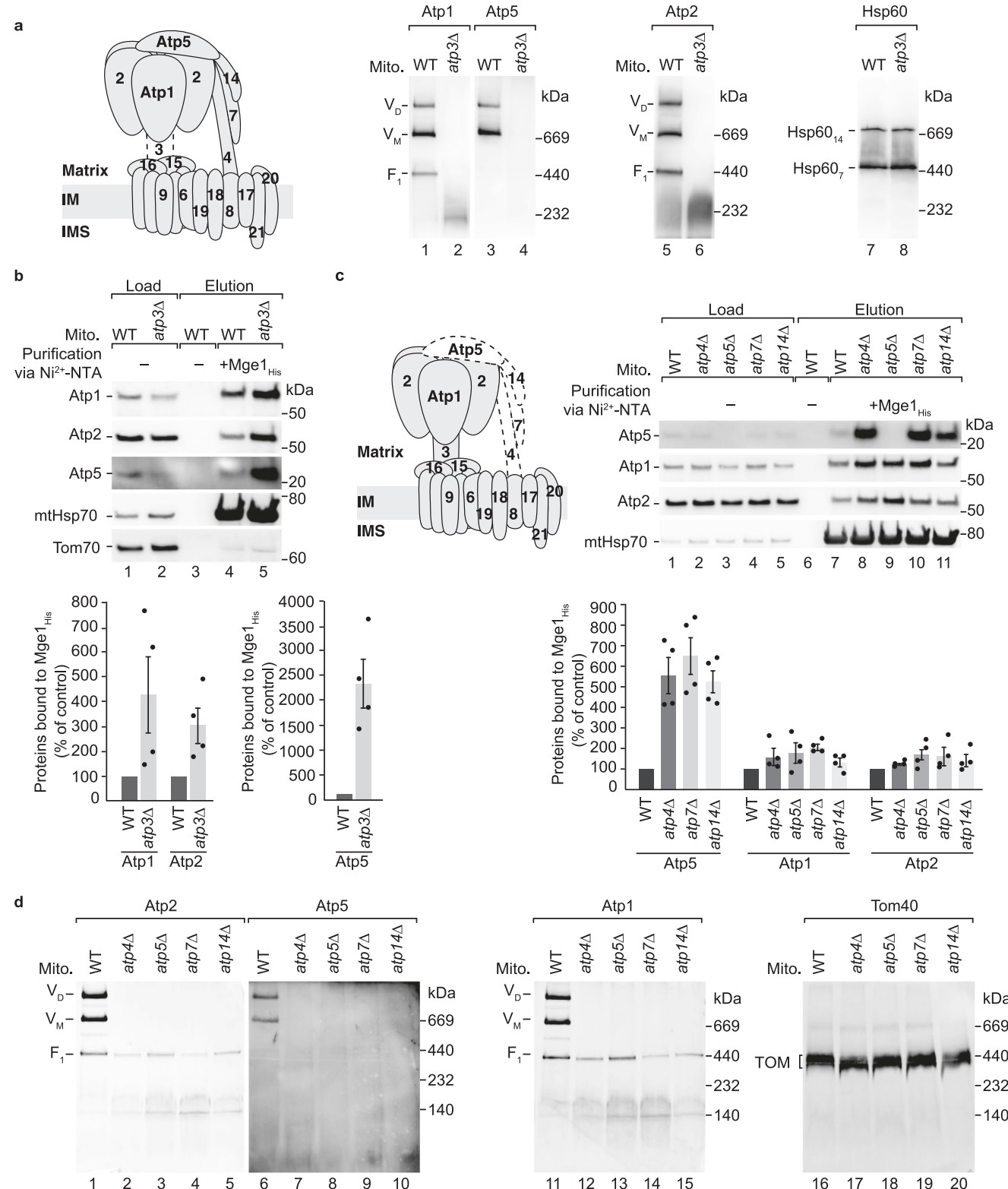

mitochondria (Supplementary Fig. 3). We conclude that mtHsp70 promotes the formation of the $F_1$ domain.

## mtHsp70 cooperates with Atp11 and Atp12

We asked whether mtHsp70 interplays with the known biogenesis factors, Atp11 and Atp12, in the assembly of the ATP synthase. Atp11 and Atp12 are critical for the formation of the $F_1$ domain. Here, Atp11 and Atp12 bind to Atp2 and Atp1, respectively, and thereby prevent the aggregation of the ATP synthase subunits[49–51]. We co-purified Atp11 and

Atp12 via mtHsp70$_{His}$ and via the Mge1$_{His}$ affinity matrix (Fig. 1b–d and 4a), demonstrating that mtHsp70 interacts with Atp11 and Atp12. The assembly of Atp1 and Atp2 was reduced in *ssc1-62* mitochondria (Fig. 3b). Therefore, we analysed whether the binding of mtHsp70 to these assembly factors and subunits of the ATP synthase was affected in *ssc1-62* mutant mitochondria using the Mge1$_{His}$ affinity matrix. The mutated allele of mtHsp70 was co-purified along with Mge1$_{His}$ (Figs. 4a, lane 5)[28], whereas the binding of Atp12 as well as to the ATP synthase subunits Atp1, Atp2 and Atp5 was strongly reduced (Fig. 4a, lane 5).

**Fig. 2 | MtHsp70 interacts with unassembled subunits of the ATP synthase.**
**a** Wild-type (WT) and *atp3Δ* mitochondria were analysed by blue native electrophoresis and immunodetection with the indicated antisera. $V_D$, dimer of the ATP synthase, $V_M$, monomer of the ATP synthase, $F_1$, $F_1$ domain. **b** Upper panel, WT and *atp3Δ* mitochondria were subjected to affinity purification via Mge1$_{His}$ coated Ni-NTA agarose. Proteins were analysed by SDS-PAGE and immunodetection with the indicated antisera. Load: 1%, elution: 100%. Lower panels, Quantification of the co-purified Atp1, Atp2 and Atp5 with Mge1$_{His}$. Depicted are mean values ± SEM of 4 independent experiments. The amount of Atp1, Atp2 or Atp5 co-eluted with mtHsp70 in WT mitochondria were set to 100% (control). Source data are provided as a Source Data file. **c** Upper panel, WT and *atp4Δ, atp5Δ, atp7Δ and atp14Δ*

mitochondria were subjected to affinity purification via Mge1$_{His}$-coated Ni-NTA agarose. Proteins were analysed by SDS-PAGE and immunodetection with the indicated antisera. Load: 1%, elution: 100%. Lower panel, Quantification of the co-purified Atp1, Atp2 and Atp5 with Mge1$_{His}$. Depicted are mean values ± SEM of 4 independent experiments. The amount of Atp1, Atp2 or Atp5 co-eluted with mtHsp70 in WT mitochondria were set to 100% (control). Source data are provided as a Source Data file. **d** WT and *atp4Δ, atp5Δ, atp7Δ and atp14Δ* mitochondria were analysed by blue native electrophoresis and immunodetection with the indicated antisera. $V_D$, dimer of the ATP synthase, $V_M$, monomer of the ATP synthase, $F_1$, $F_1$ domain.

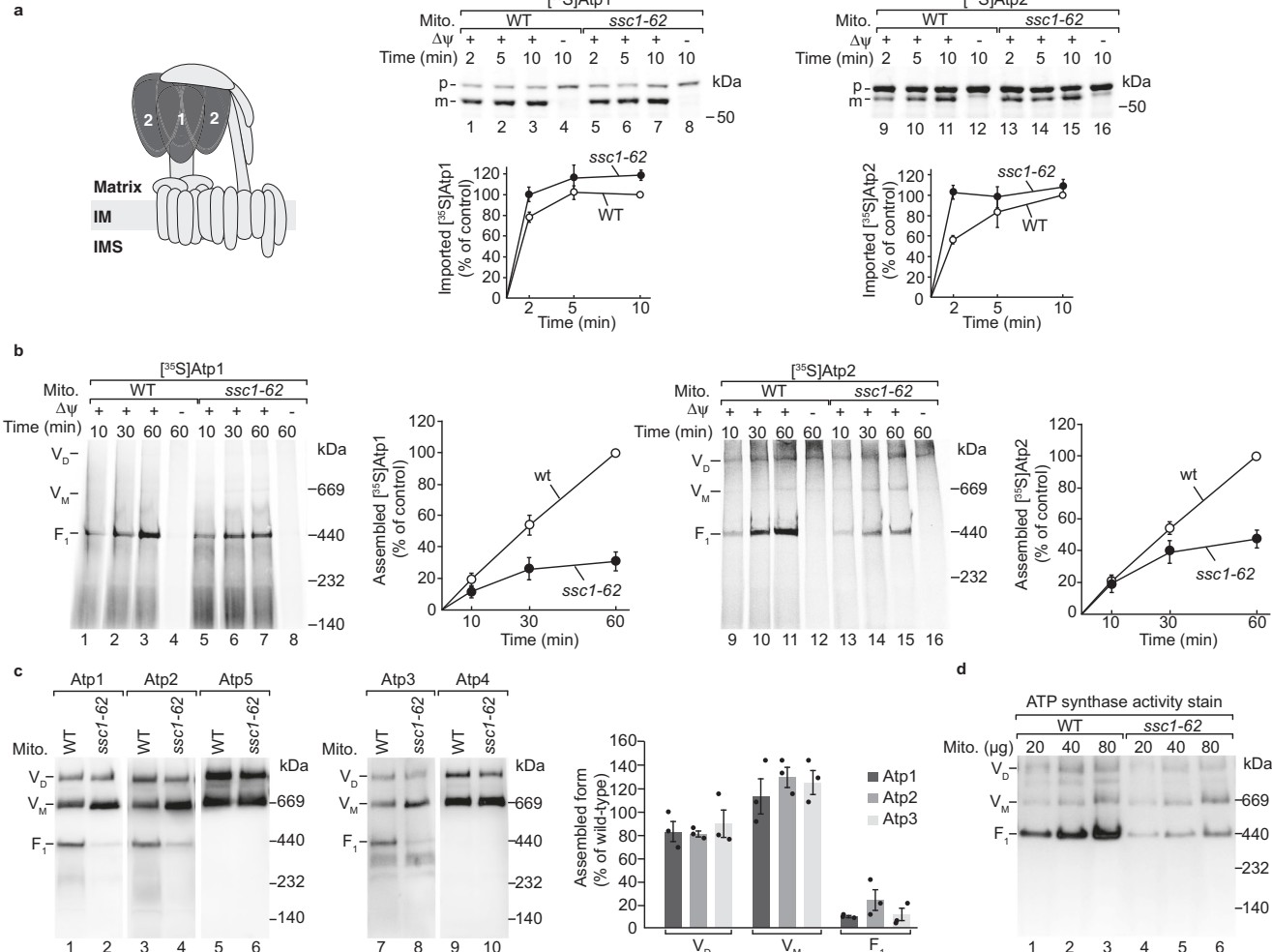

**Fig. 3 | mtHsp70 facilitates the assembly of the $F_1$ domain. a** Upper panels, [$^{35}$S]Atp1 or [$^{35}$S]Atp2 were imported into WT and *ssc1-62* mitochondria from non-stressed cells for the indicated time points. In control reactions, the membrane potential (Δψ) was depleted. Samples were analysed via SDS-PAGE and autoradiography. p, precursor, m, mature band. Lower panels: Quantification of the imported Atp1 or Atp2. Depicted are mean values ± SEM of 3 independent experiments for Atp1 and mean values ± SEM of 4 independent experiments for Atp2. The amounts of mature Atp1 or Atp2 in WT mitochondria in the longest import time point were set to 100% (control). Source data are provided as a Source Data file. **b** Upper panels, [$^{35}$S]Atp1 or [$^{35}$S]Atp2 were imported into WT and *ssc1-62* mitochondria from non-stressed cells for the indicated time points. In control reactions, the membrane potential (Δψ) was depleted. Samples were analysed via blue native electrophoresis and autoradiography. $V_D$, dimer of the ATP synthase,

$V_M$, monomer of the ATP synthase, $F_1$, $F_1$ domain. Lower panels: Quantification of Atp1 or Atp2 assembled into the $F_1$ domain. Depicted are mean values ± SEM of 3 independent experiments. The amounts of assembled Atp1 or Atp2 in WT mitochondria at the longest import time point were set to 100% (control). Source data are provided as a Source Data file. **c, d** Wild-type (WT) and *ssc1-62* mitochondria were isolated from cells that were shifted to non-permissive growth conditions. Mitochondrial proteins were analysed by blue native electrophoresis followed by either immunodetection with the indicated antisera (**c**) or by in-gel activity stain (**d**). $V_D$, dimer of the ATP synthase, $V_M$, monomer of the ATP synthase, $F_1$, $F_1$ domain. **c** Right panel, The amounts of the dimer, monomer and $F_1$ domain in WT and *ssc1-62* was determined with the indicated antisera. The formation of the forms in WT was set to 100% (control). Depicted are mean values ± SEM of 3 independent experiments. Source data are provided as a Source Data file.

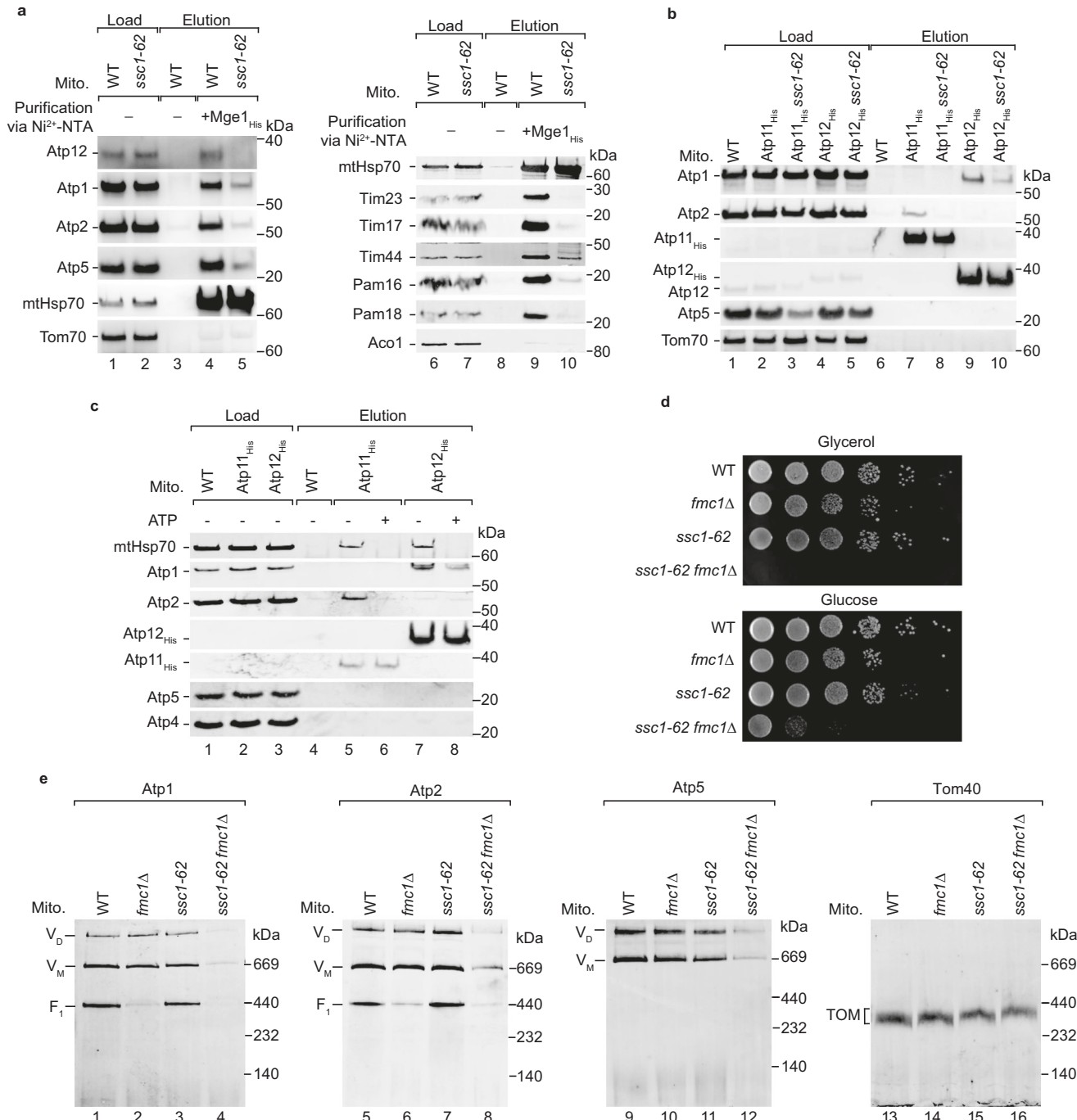

**Fig. 4 | mtHsp70 cooperates with Atp11 and Atp12 in the assembly of the F₁ domain. a** Wild-type (WT) and *ssc1-62* mitochondria from non-stressed cells were subjected to affinity purification via Mge1$_{His}$-coated Ni-NTA agarose. Proteins were analysed by SDS-PAGE and immunodetection with the indicated antisera. The blots from the left and right panels were from two independent experiments. Load: 1% (left panel) or 3% (right panel), elution: 100%. **b** WT, Atp11$_{His}$, Atp11$_{His}$ *ssc1-62*, Atp12$_{His}$, and Atp12$_{His}$ *ssc1-62* were shifted to non-permissive growth conditions. Isolated mitochondria were subjected to affinity purification via Ni-NTA agarose. Proteins were analysed by SDS-PAGE and immunodetection with the indicated antisera. Load: 1%, elution: 100%. **c** Lysed WT, Atp11$_{His}$ and Atp12$_{His}$ mitochondria were pre-incubated with or without ATP and subsequently subjected to affinity purification via Ni-NTA agarose. Proteins were analysed by SDS-PAGE and immunodetection with the indicated antisera. Load: 0.2%, elution: 100%. **d** Serial dilutions of the indicated yeast strains were grown on full medium containing either glycerol or glucose as carbon source grown at 23 °C. **e** WT, *fmc1Δ*, *ssc1-62*, and *fmc1Δ ssc1-62* mitochondria from non-stressed cells were analysed by blue native electrophoresis and immunodetection with the indicated antisera.

Similarly, the binding of TIM23 and PAM subunits to Mge1$_{His}$ affinity matrix was diminished in the *ssc1-62* mutant mitochondria (Fig. 4a, lane 10). To investigate whether mtHsp70 affects the binding of Atp1 and Atp2 to their assembly factors, we expressed His-tagged Atp11 and Atp12 in the *ssc1-62* background and grow the cells under non-permissive conditions. We found in affinity purifications that binding

of Atp1 and Atp2 to their cognate assembly factor was strongly decreased in *ssc1-62* mitochondria (Fig. 4b, lanes 8 and 10). mtHsp70 binds to client proteins in an ATP-sensitive manner[11,13,30–32]. We found that the co-purification of Atp2 with Atp11$_{His}$ was blocked and the binding of Atp1 to Atp12$_{His}$ was reduced upon preincubation of mito-chondrial lysates with ATP (Fig. 4c, lanes 6 and 8). Similarly, the

binding of mtHsp70 to Atp11 and Atp12 was blocked in the presence of ATP (Fig. 4C, lanes 6 and 8). Thus, mtHsp70 stabilises the binding of Atp1 and Atp2 to their assembly factors.

To investigate the physiological role of this interplay, we studied the genetic interaction of *ssc1-62* with either *ATP11* or *ATP12*. However, cells lacking either *ATP11* or *ATP12* do not form an ATP synthase and are therefore not capable to grow in respiratory conditions[49–51]. To circumvent this problem, we followed two independent strategies. First, we analysed *ssc1-62* strains expressing His-tagged version of either Atp11 or Atp12. We found that the cell growth and the formation of the ATP synthase are impaired in these double mutant strains (Supplementary Fig. 3b), pointing to a synthetic effect on the formation of the ATP synthase. Second, we analysed a genetic interaction of *ssc1-62* with the deletion of Fmc1. Loss of Fmc1 leads to decreased steady-state levels of Atp12 and aggregation of Atp1 and Atp2 under heat shock conditions[52]. We deleted *FMC1* in the *ssc1-62* mutant to study the interplay of mtHsp70 with Fmc1. While the single mutant strains displayed only a mild growth defect (Fig. 4d) as reported[28,52], the growth of *fmc1Δ ssc1-62* double mutant was blocked on a non-fermentable carbon source and strongly reduced on fermentable carbon source (Fig. 4d). The levels of the ATP synthase were strongly diminished in the double mutant mitochondria when the cells were grown in galactose-containing medium (Fig. 4e). Particularly, the levels of the $F_1$ domain were strongly decreased in these mutant mitochondria (Fig. 4e, lanes 4 and 8). Altogether, mtHsp70 cooperates with Atp11 and Atp12 in the formation of the $F_1$ domain.

## mtHsp70 provides Atp5 for the assembly of the ATP synthase

Atp5 strongly accumulates at mtHsp70 when the formation of the $F_1$ domain or of the peripheral stalk was blocked (Figs. 1 and 2). Atp5 links the catalytic head of the $F_1$ domain with the peripheral stalk[36] and is therefore critical for the formation of the ATP synthase. INAC promotes the formation of the peripheral stalk and its assembly into the ATP synthase. However, the mature ATP synthase is formed in the absence of INAC[53], pointing to the existence of further assembly factors. The steady-state levels of the ATP synthase subunits, mtHsp70 and Tom40 were largely unaffected in *ina17Δ* and *ina22Δ* mitochondria (Supplementary Fig. 5a), while the ATP synthase complex was mildly destabilised (Supplementary Fig. 5b)[53]. We analysed whether loss of INAC affects binding of Atp5 to mtHsp70. Excitingly, Atp5 was strongly enriched after purification via Mge1$_{His}$ affinity matrix in *ina17Δ* and *ina22Δ* mitochondria, while Atp1 and Atp2 were only mildly enriched in the mutant mitochondria (Fig. 5a, lanes 6 and 7). We asked whether mtHsp70 affects the transfer of subunits of the ATP synthase to INAC. INAC binds to the inner membrane anchor Atp4 and Atp5 of the peripheral stalk and Atp1 and Atp2 of the $F_1$ domain (Fig. 5b, lanes 5 and 7)[53]. The co-purification of Atp5, Atp1 and Atp2 along with His-tagged Ina17 and Ina22 was impaired upon preincubation of mitochondrial lysates with ATP, whereas co-purification of the membrane-anchored Atp4 of the peripheral stalk with Ina17$_{His}$ or Ina22$_{His}$ was unchanged (Fig. 5b, lane 6 and 8). Furthermore, INAC-bound Atp1, Atp2 and Atp5 can be eluted by addition of ATP (Fig. 5c, lanes 5 and 6). Since no direct interaction of INAC to mtHsp70 was observed[29,53], we propose that mtHsp70 could transiently bind via Atp5 to the assembly intermediate at INAC. The binding of Atp5 could be critical to stabilise the association of the soluble Atp1 and Atp2, but not of the membrane-anchored Atp4 to INAC. To reveal that mtHsp70 and INAC cooperate in the assembly of the ATP synthase, we deleted *INA17* or *INA22* in the *ssc1-62* background. The single deletion strains *ina17Δ* and *ina22Δ* do not show any growth defects, whereas their deletion in the *ssc1-62* leads to an impaired growth of the strains on a fermentable carbon source and lethality under respiratory growth conditions (Fig. 5d). The steady-state levels of several subunits of the ATP synthase were reduced in the double mutant mitochondria isolated from cells grown in galactose-containing medium (Supplementary Fig. 5c). The formation of the monomer and dimer of the ATP synthase was blocked in the double mutant mitochondria, while the $F_1$-domain was still present in these mutant mitochondria (Fig. 5e, lanes 4, 8 and 12). We conclude that mtHsp70 and INAC cooperate to assemble Atp5, which links the peripheral stalk with the $F_1$ domain in the mature ATP synthase.

## mtHsp70 monitors the assembly of Atp5

We analysed the assembly of $^{35}$S-labelled Atp5 in *ssc1-62* mitochondria isolated from cells grown under permissive conditions. The import of Atp5 was unaffected (Fig. 6a), whereas its assembly into the monomer and dimer of the ATP synthase was strongly accelerated (Fig. 6b, lanes 5–7). For comparison, the assembly of Atp4 was not affected and the assembly of Atp14 was mildly stimulated in the mutant mitochondria (Fig. 6a, b). Since the mutated mtHsp70 poorly interacts with Atp5 (Fig. 4a), we speculated that mtHsp70 could function as a quality control factor in the assembly of the ATP synthase. To test this idea, we searched for Atp5 mutants that were defective in the assembly of the ATP synthase. Based on the cryo-EM structure of the yeast ATP synthase, we exchanged glycine 183 to alanine at the interface between Atp5 and the catalytic head[36] (previously, the corresponding amino acid was also termed Atp5$^{G166}$ according to the amino acid position in the processed protein[63]). Imported [$^{35}$S]Atp5$^{G183A}$ fails to assemble efficiently in wild-type mitochondria in vitro (Fig. 6c). Strikingly, in *ssc1-62* mitochondria the Atp5 variant assembles efficiently into the mature monomeric and dimeric ATP synthase (Fig. 6d). To test whether this phenomenon can be generally observed, we searched for mutants at the interface of Atp5 to the $F_1$ domain. Previously, two phosphosites (serine 48 and threonine 139) of Atp5 were reported[64], from which threonine 139 locates at the interface to Atp1[36]. We exchanged both, the serine and threonine residues, to either alanine (Atp5$^{S48A\ T139A}$) or glutamate (Atp5$^{S48E\ T139E}$) and analysed the assembly of the $^{35}$S-labelled Atp5 variants on blue native electrophoresis. The assembly of both Atp5 variants were impaired in wild-type mitochondria and stimulated in *ssc1-62* mitochondria (Supplementary Fig. 6). The observations indicate that binding to mtHsp70 is critical to prevent unproductive assembly of Atp5 variants.

To challenge the potential role of mtHsp70 in the control of the assembly of the ATP synthase, we expressed Atp5 and Atp5$^{G183A}$ in wild-type and *ssc1-62* cells under control of a galactose-inducible promoter, which results in mild overexpression of either Atp5 or Atp5$^{G183A}$. Using the Mge1$_{His}$ affinity matrix, we found that the excess of Atp5 and particularly of the Atp5$^{G183A}$ variant binds to mtHsp70 in wild-type mitochondria but not in *ssc1-62* mitochondria (Fig. 7a). Growth analysis revealed that the overexpression of Atp5 or Atp5$^{G183A}$ was toxic in the *ssc1-62* strain but not in wild-type cells (Fig. 7b), indicating that binding of free Atp5 to mtHsp70 is critical for cell growth. We wondered whether the stability of non-assembled Atp5 was affected in *ssc1-62* mitochondria. Therefore, we expressed HA-tagged Atp5 and Atp5$^{G183A}$ in wild-type and *ssc1-62* cells. The HA-tag enables us to detect the additional Atp5 copies. We found an increased accumulation of the HA-tagged Atp5 variants in the mutant mitochondria compared to wild-type mitochondria (Fig. 7c, lanes 7, 8, 11, and 12), indicating that their degradation is impaired in the *ssc1-62* mutant. MtHsp70 cooperate with the AAA ATPase Pim1 in protein degradation in the matrix[65–68]. Indeed, we found stabilisation of the HA-tagged Atp5 variants in the *pim1Δ* compared to wild-type cells (Fig. 7d), while the accumulation was not increased when Yta12 of the inner membrane m-AAA ATPase was deleted (Fig. 7d). These observations point to cooperation of mtHsp70 and Pim1 in the removal of unassembled Atp5 variants. Altogether, we conclude that mtHsp70 controls the assembly of Atp5 and thereby balances the formation of the ATP synthase.

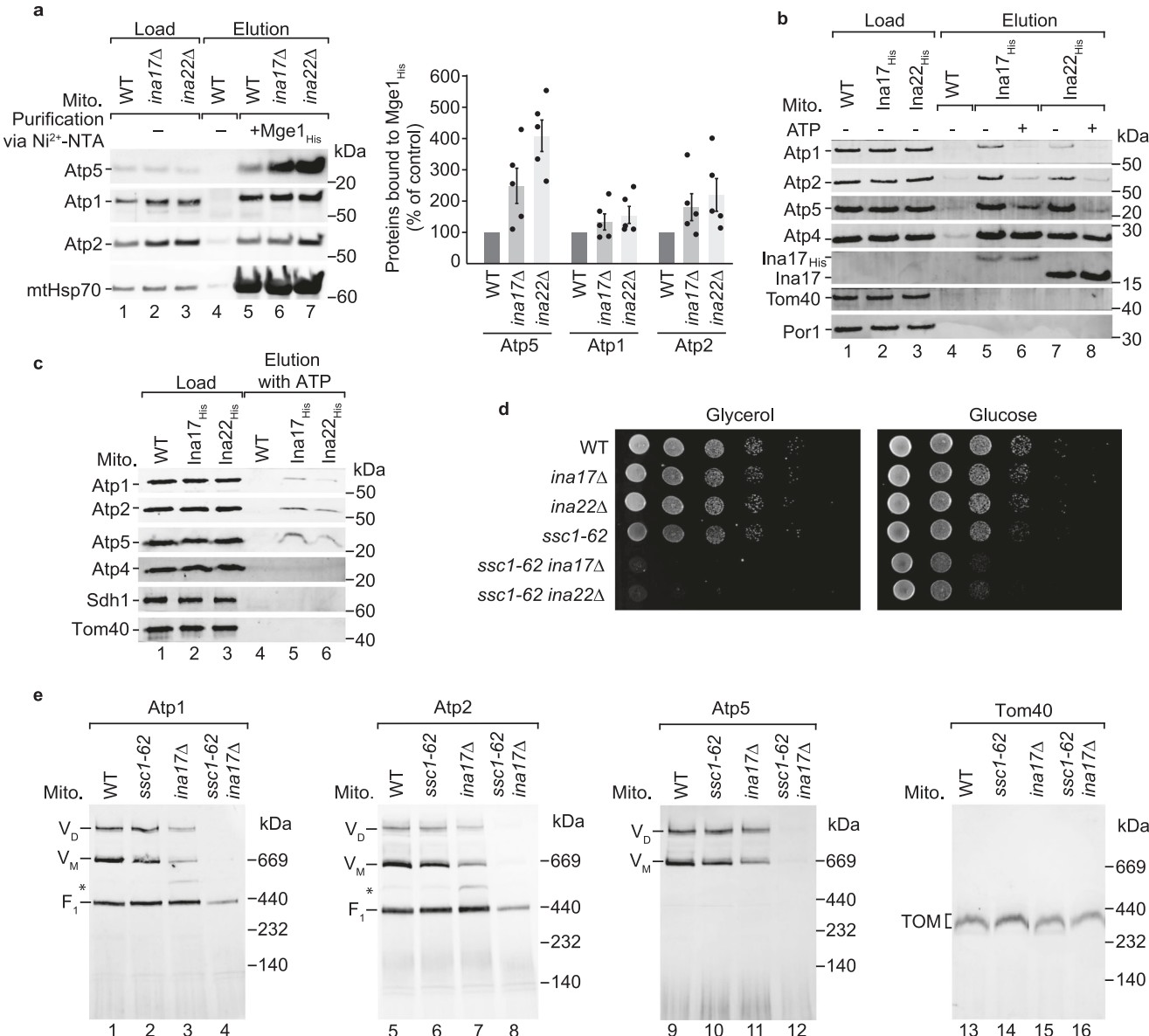

**Fig. 5 | mtHsp70 cooperates with INAC to link the $F_1$ domain to the peripheral stalk. a** Left panel, Wild-type (WT), *ina17Δ* and *ina22Δ* mitochondria were subjected to affinity purification via Mge1$_{His}$ coated Ni-NTA agarose. Proteins were analysed by SDS-PAGE and immunodetection with the indicated antisera. Load: 1%, elution: 100%. Right panel, Quantification of Atp1, Atp2 and Atp5 co-purified with Mge1$_{His}$. Depicted are mean values ± SEM of 5 independent experiments. The amount of Atp1, Atp2 or Atp5 co-eluted with mtHsp70 in WT mitochondria was set to 100% (control). Source data are provided as a Source Data file. **b** Lysed WT, Ina17$_{His}$ and Ina22$_{His}$ mitochondria were pre-incubated with or without ATP and subsequently subjected to affinity purification via Ni-NTA agarose. Proteins were analysed by SDS-PAGE and immunodetection with the indicated antisera. Load: 0.5%, elution: 100%. **c** WT, Ina17$_{His}$ and Ina22$_{His}$ mitochondria were subjected to affinity purification via Ni-NTA agarose. Where indicated, bound proteins were eluted by incubation with ATP. Proteins were analysed by SDS-PAGE and immunodetection with the indicated antisera. Load: 1%, elution: 100%. **d** Serial dilutions of the indicated yeast strains were grown on full medium containing either glycerol or glucose as carbon source grown at 23 °C. **e** WT, *ina17Δ*, *ssc1-62*, and *ina17Δ ssc1-62* mitochondria from non-stressed cells were analysed by blue native electrophoresis and immunodetection with the indicated antisera. Asterisk marks unknown Atp1 and Atp2-containing protein complex in *ina17Δ* mitochondria.

## Discussion

The ATP synthase is essential for ATP production and cellular energy metabolism. Defects of the ATP synthase and its assembly have been linked to various diseases[69–71]. Nevertheless, the assembly of the $F_1$ domain and the peripheral stalk of the mitochondrial ATP synthase is poorly understood. We report here that mtHsp70 fulfils critical steps in the assembly process (Fig. 7e). First, the chaperone cooperates with Atp11 and Atp12 to form the $F_1$ domain. Second, mtHsp70 delivers Atp5 to INAC to allow assembly of the peripheral stalk with the $F_1$ domain. Inactivation of the mtHsp70 in the absence of INAC blocks the assembly of the ATP synthase, indicating that mtHsp70 is a so far missing factor

that promotes biogenesis of soluble modules of the ATP synthase. In this process, mtHsp70 also acts as quality control factor to minimise the integration of assembly-defective Atp5 variants into the ATP synthase.

mtHsp70 plays a dual role in the biogenesis of Atp1 and Atp2. First, it promotes import of both precursor proteins via the TIM23 pathway[16,20,72]. Second, mtHsp70 further stabilises the interaction of Atp1 and Atp2 to Atp12 and Atp11, respectively, to promote subsequent assembly steps. Loss of either Atp11 or Atp12 causes aggregation of Atp1 and Atp2 and blocks formation of the $F_1$ domain[73]. Therefore, binding to the different factors likely maintains Atp1 and Atp2 in an assembly-competent state to allow formation of the $F_1$ domain.

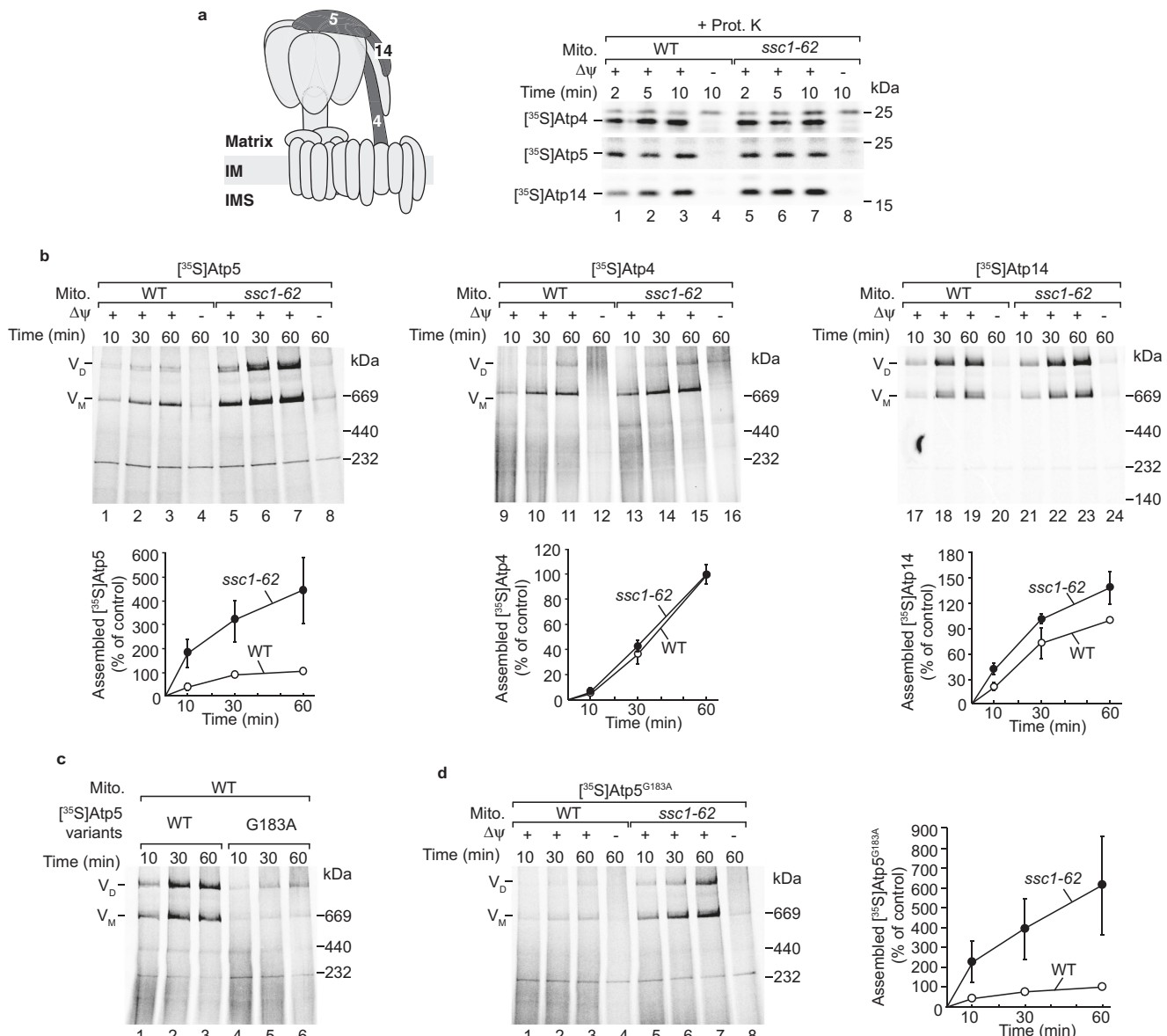

**Fig. 6 | mtHsp70 controls the assembly of Atp5. a** [³⁵S]Atp4, [³⁵S]Atp5, or [³⁵S]Atp14 were imported into wild-type (WT) and *ssc1-62* mitochondria from non-stressed cells for the indicated time period. Non-imported precursor proteins were removed by proteinase K (Prot. K). In control reactions, the membrane potential (Δψ) was depleted. Samples were analysed via SDS-PAGE and autoradiography. p, precursor, m, mature band. **b** Upper panels, [³⁵S]Atp5, [³⁵S]Atp4 or [³⁵S]Atp14 were imported into WT and *ssc1-62* mitochondria from non-stressed cells for the indicated time periods. In control reactions, the membrane potential (Δψ) was depleted. Samples were analysed via blue native electrophoresis and autoradiography. $V_D$, dimer of the ATP synthase, $V_M$, monomer of the ATP synthase. Lower panels: Quantification of Atp5, Atp4 or Atp14 assembled into the dimer of the ATP synthase. Depicted are mean values ± SEM of 3 independent experiments for Atp4 and

Atp14, and mean values ± SEM of 4 independent experiments for Atp5. The amounts of Atp5, Atp4 or Atp14 assembled into the ATP synthase dimer in WT mitochondria at the longest import time point were set to 100% (control). Source data are provided as a Source Data file. **c, d** [³⁵S]Atp5 (**c**) or [³⁵S]Atp5^G183A (**d**) were imported into WT and *ssc1-62* mitochondria isolated from non-stressed cells for the indicated time periods. In control reactions, the membrane potential (Δψ) was depleted. Samples were analysed via blue native electrophoresis and auto-radiography. $V_D$, dimer of the ATP synthase, $V_M$, monomer of the ATP synthase. **d** Right panel: quantification of Atp5^G183A assembled into the monomer and dimer of the ATP synthase. Depicted are mean values ± SEM of 4 independent experiments. The amounts of assembled Atp5^G183A in WT mitochondria at the longest import time point were set to 100% (control). Source data are provided as a Source Data file.

MtHsp70 also delivers Atp5 into the assembly line to link the $F_1$ domain to the peripheral stalk. Here, it cooperates with INAC which appears to form a scaffold to facilitate assembly of the modules of the ATP synthase[53,54]. Remarkably, inactivation of both INAC and mtHsp70 blocks formation of the ATP synthase, indicating that both protein factors play a partly overlapping but crucial function in the linkage of the $F_1$ domain with the peripheral stalk. So far, an assembly intermediate of the peripheral stalk was reported that lacks Atp5[48]. We propose that mtHsp70 delivers Atp5 to such assembly intermediate to allow the formation of the ATP synthase. A substantial fraction of Atp5 is bound to

mtHsp70 in wild-type cells. This observation is supported by the higher copy numbers of Atp5 compared to other subunits of the peripheral stalk in mitochondria (Supplementary Fig. 7)[1]. Furthermore, Atp5 accumulates at mtHsp70 when its assembly is impaired. One can speculate that mtHsp70 provides a reservoir of unassembled Atp5 that is delivered into the assembly line of the ATP synthase when needed. Similarly, other soluble ATP synthase subunits like Atp1 and Atp2 accumulate at mtHsp70 in mutants defective in forming the ATP synthase.

MtHsp70 not only function here as assembly factor, but also monitors the integration of Atp5 into the ATP synthase. In *ssc1-62*

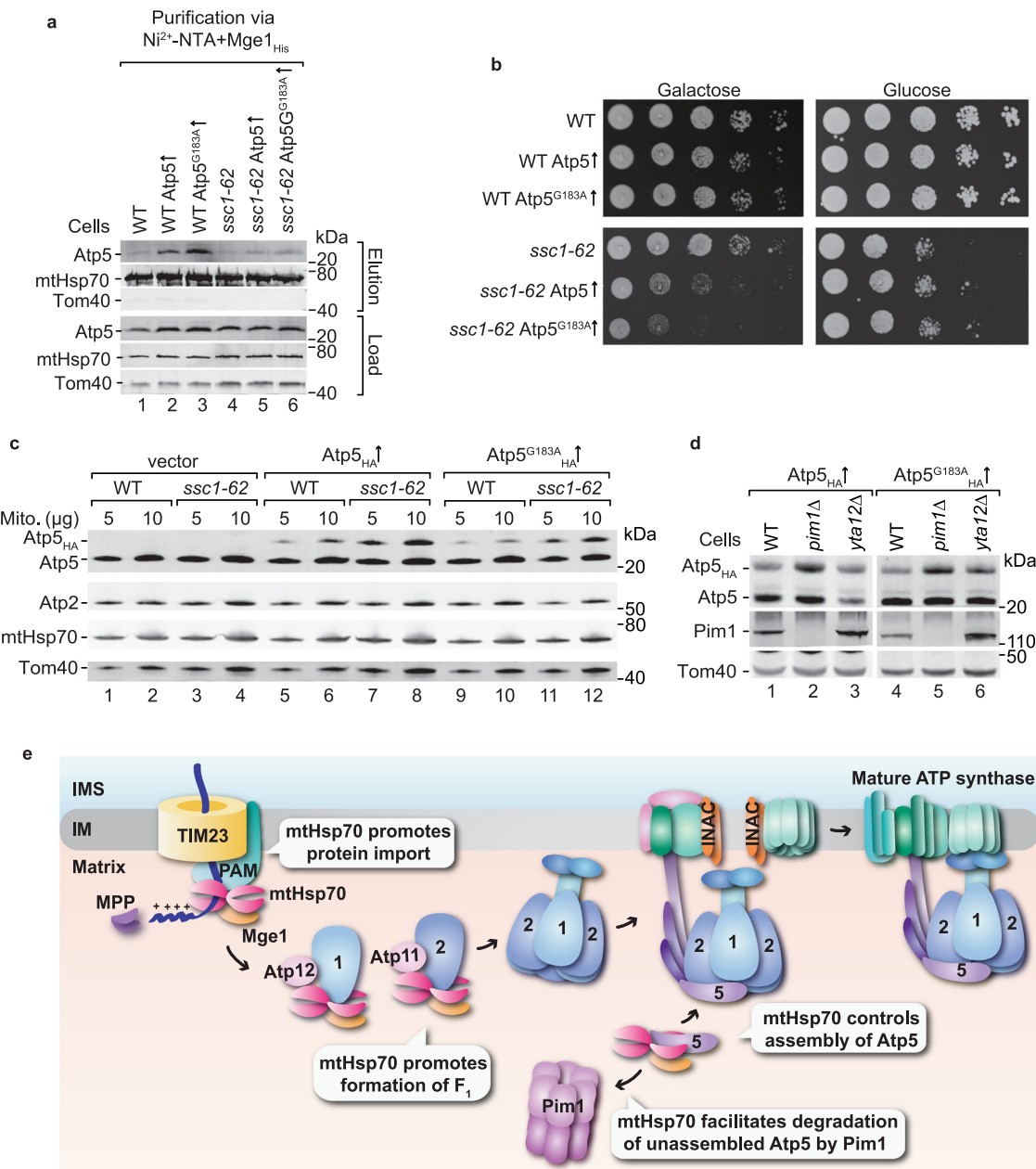

**Fig. 7 | mtHsp70 and Pim1 promotes degradation of unassembled Atp5. a** Wild-type (WT) and *ssc1-62* cells with or without overexpression of either Atp5 or Atp5$^{G183A}$ were grown under permissive conditions and subjected to affinity purification via Mge1$_{His}$ coated Ni-NTA agarose. Proteins were analysed by SDS-PAGE and immunodetection with the indicated antisera. Load: 1%, elution: 100%. **b** Serial dilutions of the indicated yeast strains were grown on selective medium containing either glucose or galactose as carbon source grown at 33 °C. The overexpression of Atp5 and Atp5$^{G183A}$ is induced upon growth on galactose. **c** Mitochondria isolated from non-stressed WT and *ssc1-62* cells expressing HA-tagged Atp5 or Atp5$^{G183A}$ were lysed under denaturing conditions and proteins were analysed with the indicated antisera. **d** Total cell extracts from WT, *pim1Δ*, or *yta12Δ* strains expressing HA-tagged Atp5 or Atp5$^{G183A}$ were lysed under denaturing conditions and proteins were analysed with the indicated antisera. **e** Hypothetical model of the role of mtHsp70 in the formation of the ATP synthase. MtHsp70 promotes import of ATP synthase subunits into the mitochondrial matrix. The chaperone cooperates with Atp11 and Atp12 to assemble Atp2 and Atp1 subunits into the F$_1$ domain. MtHsp70 delivers Atp5 to the assembly line to link the peripheral stalk and the F$_1$ domain. Unassembled Atp5 can be degraded by Pim1.

mutant, the binding of Atp5 to mtHsp70 and the control of the assembly of Atp5 were compromised. We found that Atp5 variants can efficiently assemble into the ATP synthase in the mutant mitochondria, while its assembly is impaired in wild-type mitochondria. Furthermore, overexpression of wild-type Atp5 and an assembly-defective variant is toxic for the cell when mtHsp70 is not fully functional. We conclude that mtHsp70 acts as quality control factor that balances the formation of the ATP synthase. MtHsp70 delivers unassembled Atp5 variants for degradation by Pim1. The AAA ATPase Pim1 plays a central role to prevent the accumulation of misfolded and damaged proteins[33,65–68]. Previously, it was reported that Pim1 degrades Atp1, Atp2 and Atp7[65,67], reflecting an important role of Pim1 in the degradation of unassembled subunits of the ATP synthase. Pim1 is known to cooperate with mtHsp70 in the degradation of misfolded proteins[74], supporting our observation that overexpressed Atp5 is stabilised in both *pim1Δ* and *ssc1-62* mutants.

MtHsp70 is a well-established factor for protein import and protein folding in mitochondria. Our analysis uncovered an important role of

the chaperone in the formation of the ATP synthase. Similarly, mtHsp70 plays a critical role in the formation of complex IV and Hsp60 rings[26–29]. Thus, mtHsp70 is an important assembly and quality control factor in the formation of multi-subunit protein machinery in mitochondria.

## Methods

### Yeast strains and growth conditions

The yeast strains used in this study are listed in Supplementary Table 1. YPH499, mtHsp70$_{His}$, *ssc1-62* and its corresponding wild-type strain have been described before[28,75]. The strains *atp3Δ, atp4Δ, atp5Δ, atp7Δ, atp14Δ, ina17Δ, ina22Δ, pim1Δ* and *yta12Δ* and their corresponding wild-type BY4741 were purchased from Euroscarf. For the generation of the Atp1$_{His}$ strain, the sequence for Deca-His-tag was chromosomally inserted before the stop codon of the *ATP1* open reading frame using a *HIS3MX6* cassette for homologous recombination[76]. For generation of the Atp11$_{His}$, Atp12$_{His}$, Ina17$_{His}$, Ina22$_{His}$, Atp11$_{His}$ *ssc1-62*, Atp12$_{His}$ *ssc1-62*, Ina17$_{His}$ *ssc1-62*, Ina22$_{His}$ *ssc1-62* strains, the sequence for a deca-His-tag was chromosomally inserted before the stop codon of the respective open reading frame using a hygromycin selection marker for homologous recombination. For the generation of *ssc1-62 atp11Δ, ssc1-62 atp12Δ, ssc1-62 ina17Δ, ssc1-62 ina22Δ* and *ssc1-62 fmc1Δ*, the respective open reading frame of *ATP11, ATP12, INA17, INA22* or *FMC1* was chromosomally deleted in a Ssc1 shuffle strain using a *HIS3MX6* selection marker for homologous recombination. The *ssc1-62* strain was generated by plasmid shuffling[28]. The Ssc1 shuffle strain contains a pYEp352 plasmid encoding for the open reading frame of *SSC1* and a *URA3* selection marker. The chromosomal *SSC1* locus was disrupted with a *ADE2* marker. These strains were transformed with a pFL39 plasmid containing a *TRP* marker and the genetic information for either the mutant allele *SSC1-62* or for wild-type *SSC1*. Subsequently, the deletion strains were cultured on media containing 5-fluoroorotic acid to induce loss of the pYEp352 plasmid encoding for wild-type *SSC1* utilising the *URA3* marker. For overexpression of *ATP5* and its variant, a pRS415 plasmid encoding either untagged or HA-tagged *ATP5, and ATP5$^{G183A}$* variant under the control of *GAL* promoter was transformed into *ssc1-62* or *pim1Δ* and the corresponding wild-type strain. Yeast cells were cultured with YP (1% [w/v] yeast extract; 2% [w/v] bacto-peptone) containing 2% glucose [w/v], 2% [w/v] sucrose or 3% glycerol [v/v] at 23–30 °C. For heat shock, *ssc1-62* cells were shifted to 37 °C for 32–36 h. For overexpression of *ATP5* and its variants, yeast cells were cultured with selective medium at 33 °C (0.67% [w/v] yeast nitrogen base, 0.07% [w/v] amino acid mixture lacking leucine, 2% [w/v] galactose and 0.1% [w/v] raffinose).

### Plasmids construction

The plasmids and primers that are used in this study are listed in Supplementary Tables 2 and 3. For overexpression of *ATP5*, the open reading frame of *ATP5* plus 1000 bp upstream and 576 bp downstream was amplified by KOD hot start DNA polymerase (Sigma Aldrich) using genomic DNA as template. The fragment was inserted into a pRS415 with quick ligase (New England Biolab). The *GAL1* promoter was amplified by KOD hot start DNA polymerase (Sigma Aldrich) using pYES2 as template. Subsequently, the *GAL1* promoter was inserted in front of the start codon of *ATP5* replacing the 1000 bp upstream of the open reading frame of *ATP5*. A fragment encoding for a *ATP5* fused to triple HA-tag were inserted into pRS415 background under the control of a *GAL1* promoter.

### Mitochondria isolation

Yeast cells were cultured in full media or minimal media until the early logarithmic growth phase. Subsequently, mitochondria were isolated by differential centrifugation[77]. To this end, cells were harvested by centrifugation (5300 *g*, 8 min, 20 °C) and washed with dH$_2$O. The wet weight of the cell pellets was determined. The cells were resuspended and incubated with DTT buffer (10 mM DTT, 100 mM Tris/H$_2$SO$_4$ pH

9.4) for 30 min at 30 °C under constant shaking to soften the cell wall. Subsequently, cells were pelleted, washed with zymolyase buffer (1.2 M sorbitol, 20 mM Tris-HCl pH 7.4) and incubated with zymolyase buffer containing 5 mg zymolyase per g wet weight for 40 min at 30 °C under constant shaking to break the cell wall. After incubation with zymolyase, cells were collected, washed with zymolyase buffer and resuspended with ice-cold lysis buffer (0.6 M sorbitol, 1 mM EDTA, 0.2% [w/v] bovine serum albumin (BSA), 1 mM phenylmethylsulfonyl fluoride (PMSF), 10 mM MOPS/KOH pH 7.4). Spheroplasts were broken by glass-Teflon homogeniser. Cell debris and nuclei were removed by centrifugation (2000 × *g*, 5 min, 4 °C). Subsequently, the supernatant was subjected to centrifugation (17,000 × *g*, 10 min, 4 °C) to pellet mitochondria. The mitochondrial pellet was washed with SEM buffer (250 mM sucrose, 1 mM EDTA, 10 mM MOPS-KOH, pH 7.2) and re-isolated by centrifugation (17,000 × *g*, 10 min, 4 °C). Mitochondria were resuspended with small amounts of SEM buffer and the protein concentration was adjusted to 10 mg/ml. Aliquots were snap-frozen by liquid nitrogen and stored at −80 °C until use.

### Affinity purifications

Mitochondria were solubilized with lysis buffer (20 mM Tris/HCl pH 7.4, 0.1 mM EDTA, 50 mM NaCl, 10% [v/v] glycerol) containing 1% [w/v] digitonin, 10 mM imidazole and 1 mM PMSF in a final concentration of 1 mg protein/ml, and incubated for 20 min at 4 °C under constant rotation. Insoluble cell debris was removed by centrifugation (17,000 × *g*, 10 min, 4 °C) and the supernatant was incubated with pre-equilibrated Ni$^{2+}$-NTA Agarose (Qiagen) for 1 h at 4 °C. Subsequently, the Ni$^{2+}$-NTA beads were washed with an excess amount of lysis buffer containing 0.1% [w/v] digitonin and 40 mM imidazole. Bound proteins were eluted by lysis buffer containing 0.1% [w/v] digitonin and 250 mM imidazole. When indicated, solubilisation were performed in the presence of 20 mM ATP pH 7.5 in lysis buffer followed by purification as described above. For elution with ATP, the Ni$^{2+}$-NTA beads were incubated with 20 mM ATP pH 7.5 in lysis buffer.

### Mge1 binding assay

To prepare the Mge1 affinity matrix, Mge1 fused to a hexa-His-tag was expressed in *Escherichia coli* BL21 strain and purified under native conditions[28]. Purified recombinant Mge1$_{His}$ were coupled to Ni$^{2+}$-NTA agarose (Qiagen) for 1 h at 4 °C. The beads were washed extensively with 500 mM NaCl, 20 mM Tris/HCl pH 8.0 and 50 mM imidazole and bound proteins were eluted with 500 mM imidazole, 50 mM NaCl and 20 mM Tris/HCl pH 6,8. For the binding assay, Mge1$_{His}$ was coupled to Ni$^{2+}$-NTA in 0.1% [w/v] digitonin in lysis buffer. After excessive washing with 0.1% [w/v] digitonin and 10 mM imidazole in lysis buffer, the Mge1$_{His}$ affinity matrix was incubated with mitochondria lysed with lysis buffer containing 1% [w/v] digitonin, 10 mM imidazole and 1 mM PMSF. All following steps were performed as described above.

### Blue native electrophoresis

Isolated mitochondria were resuspended in lysis buffer containing 1% [w/v] digitonin for 20 min on ice. After removal of insoluble material (17,000 × *g*, 10 min, 4 °C), protein complexes were separated by blue native electrophoresis[57,77]. We used self-cast blue native gradient gels (4–12% [w/v] acrylamide, 0.19–0.40% [w/v] bis-acrylamide, 67 mM ε-amino n-caproic acid, 50 mM Bis-Tris/HCl pH 7.0). The electrophoresis was performed at 4 °C, with cathode buffer (0.02% [w/v] Coomassie G250, 50 mM Tricine, 15 mM Bis-Tris/HCl pH 7.0) and anode buffer (50 mM Bis-Tris/HCl pH 7.0). For immunodetection, the cathode buffer was exchanged to clear cathode buffer lacking Coomassie G after the proteins entered the gel.

### In-gel activity stain

We used in-gel activity stain to show the activity of the ATP synthase[61,62]. Isolated mitochondria were solubilized with 1% [w/v] digitonin in lysis

buffer. Insoluble material was removed by centrifugation ($17{,}000 \times g$, 10 min, 4 °C). Solubilized proteins were separated by blue native electrophoresis as described above. The blue native gel was incubated with $dH_2O$ for 20 min, followed by an incubation in ATP buffer (5 mM $MgCl_2$ 50 mM glycine, 20 mM ATP, adjusted to pH 8.4 with NaOH) for 20 min. Subsequently, gels were treated with 10% [w/v] $CaCl_2$. Upon precipitation of calcium phosphate, the staining was stopped in $dH_2O$.

### Cell-free synthesis of $^{35}$S-labelled proteins

Precursors of the ATP synthase were labelled with [$^{35}$S]methionine in a cell-free translation system based on rabbit reticulocyte lysate. Templates containing the open reading frame and a SP6 promoter and a ribosome binding site were generated by PCR using yeast genomic DNA as template. We used the MEGAscript SP6 Transcription Kit (Thermo Fisher Scientific) to generate mRNA from the PCR products. For the in vitro translation, we used rabbit reticulocyte lysate from either the Flexi® Rabbit Reticulocyte Lysate System (Promega) or the TNT® Quick Coupled Transcription/Translation System (Promega) following the manufacturer's recommendations. The reaction was performed in the presence of [$^{35}$S]methionine for 90 min at 30 °C. Reaction was stopped by addition of 20 mM methionine. Synthesised proteins were snap-frozen by liquid nitrogen and stored in −80 °C.

### In vitro protein import

Import of $^{35}$S-labelled proteins into mitochondria was performed[77]. $^{35}$S-labelled precursor proteins were incubated with isolated mitochondria in import buffer (1% (w/v) BSA, 250 mM sucrose, 80 mM KCl, 5 mM $MgCl_2$, 10 mM MOPS/KOH pH 7.2, 5 mM methionine, 2.5 mM $KH_2PO_4$ pH 7.2, 2 mM ATP, 2 mM NADH, 120 µg/ml creatine kinase, 12 mM creatine phosphate) for the indicated time periods. Import reaction was stopped by addition of 1 µM valinomycin. Mitochondria were re-isolated ($13{,}000 \times g$, 4 °C, 10 min) and washed with ice-cold SEM Buffer. Mitochondrial proteins were analysed either by SDS-PAGE or by blue native electrophoresis followed by autoradiography using a Typhoon FLA-9000 (GE Healthcare).

### Analysis of membrane association

Mitochondrial membranes were experimentally separated from the soluble proteins following established procedures[77]. Isolated mitochondria were incubated on ice for 30 min in EM buffer (1 mM EDTA, 10 mM MOPS-KOH, pH 7.2). Subsequently, mitochondria were sonicated on ice with a bioruptor (Diogenode). Membrane fractions were isolated by centrifugation ($100{,}000 \times g$, 4 °C, 45 min). Soluble and membrane fractions were analysed via blue native electrophoresis.

### Yeast growth for mass spectrometry

For SILAC analysis, BY4741, BY4741 $\rho^0$, mtHsp70$_{His}$, and mtHsp70$_{His}$ $\rho^0$ strains were cultured with minimal medium (0.17% [w/v] bacto-yeast nitrogen base without amino acids, 0.5% [w/v] ammonium sulphate, 0.15% [w/v] amino acid mix, 2% [w/v] galactose, 0.1% [w/v] glucose). For the wild-type and mtHsp70$_{His}$ strains, the media was either supplemented with light arginine and lysine, or with medium arginine ($^{13}C_6$) and lysine ($D_4$). For $\rho^0$ and mtHsp70$_{His}$ $\rho^0$ strains, the media was supplemented with heavy arginine ($^{13}C_6{}^{15}N_4$) and lysine ($^{13}C_6{}^{15}N_2$) (Eurisotop). Mitochondria were either directly analysed by mass spectrometry or subjected for affinity purification via either mtHsp70$_{His}$ or via a Mge1$_{His}$ affinity matrix as described above.

### Mass spectrometry sample preparation

Differentially SILAC-labelled samples were combined in a 1:1 ratio. 50 µg mitochondria were used for mitochondrial protein analysis and 100% of the elution fraction from affinity purification via mtHsp70$_{His}$ or Mge1$_{His}$ were used for protein analysis. Proteins were precipitated by addition of acetone at a sample to acetone ratio of 1:4 [v/v] followed by incubation at −20 °C overnight. Subsequently, proteins

were collected by centrifugation (15 min, $20{,}000 \times g$, 4 °C) and washed twice with ice-cold acetone. Protein pellets were air-dried and resuspended in freshly prepared TEAB buffer (8 M urea, 100 mM triethylammonium bicarbonate), followed by incubation at 37 °C for 45 min under constant agitation. Subsequently, samples were incubated with 4 mM DTT at 56 °C for 30 min to break disulphide bonds. Next, samples were alkylated with 8 mM chloroacetamide for 30 min at room temperature[78]. The reaction was quenched by incubation with 4 mM DTT for 15 min at room temperature. Subsequently, samples were diluted to reach a final concentration of 1.6 M urea and 20 mM triethylammonium bicarbonate. Trypsin (Promega) was added at an enzyme-to-protein ratio of 1:100 [w/w] and incubated overnight at 37 °C. The resulting peptides were desalted using 50 mg Sep-Pak C$_{18}$ cartridges (Waters), dried using a vacuum centrifuge, resuspended in 5% [v/v] acetonitrile, and the peptide concentration was determined using the fluorometric peptide assay kit (Thermo Fisher Scientific). Subsequently, peptides were dried using a vacuum centrifuge.

### Mass spectrometry

The Orbitrap Fusion Lumos mass spectrometer coupled to an UltiMate 3000 RSLCnano UHPLC system (both Thermo Fisher Scientific) was used for mass spectrometry analysis. Peptides were resuspended in 5% [v/v] acetonitrile, 5% [v/v] formic acid. 1 µg of peptides was loaded onto a 30 cm analytical column at a flow rate of 600 nL/min using 100% solvent A (0.1% [v/v] formic acid in water). Analytical columns were produced in-house by the generation of spray tips from fused silica capillaries (360 µm outer diameter, 100 µm inner diameter) with a P-2000 laser puller (Sutter Instruments). Spray tips were packed with 3 µm ReproSil-Pur AQ C$_{18}$ particles (Dr. Maisch). Peptides were separated with 120 min (DIA measurements) and 240 min (DDA measurements) linear gradients from 5-35% [v/v] solvent B (90% [v/v] acetonitrile, 0.1% [v/v] formic acid) at a flow rate of 300 nl/min. For DDA analyses, MS1 spectra were acquired from m/z 350-1,200 in the Orbitrap mass analyser (resolution 60,000, AGC target setting of $4 \times 10^5$). For fragmentation, the most intense precursor ions were selected for fragmentation (top speed mode, 3 sec cycle time), isolated using the quadrupole with a m/z 1.6 isolation width and fragmented by higher energy collisional dissociation (HCD) with a normalised collision energy (NCE) of 27%. MS2 spectra were acquired in the Orbitrap mass analyser (resolution 30,000 dynamic exclusion: 120 sec). For DIA analyses, MS1 scans were acquired from m/z 350-1200 in the Orbitrap analyser (resolution: 120,000, maximum injection time: 20 msec, AGC target setting: $5 \times 10^5$). MS2 scans were defined to cover the DIA MS1 scan range with 36 scan windows of 24.1 m/z each, resulting in an overlap of 0.5 m/z and a cycle time of 3.44 sec. Peptide ions were fragmented by HCD with 27% NCE and MS2 spectra acquired in the Orbitrap mass analyser (resolution: 30,000, maximum injection time: 60 msec, AGC target setting: $1 \times 10^6$).

### Spectronaut analysis

Thermo*.raw files were analysed using Spectronaut 14.7.20 (Biognosys). Hybrid spectral libraries were generated from both DDA and DIA files with the Pulsar search engine integrated in Spectronaut, applying the following parameters: precursor ion mass tolerance: dynamic; Orbitrap fragment ion mass tolerance: dynamic; fixed modification: carbamidomethylation at cysteine; variable modifications: oxidation at methionine, acetylation at protein N-terminus, and deamidation at asparagine/glutamine; enzyme: trypsin; number of allowed missed cleavage sites: 2; minimum peptide length: 5 amino acids. For spectral library generation, the "Labelling Applied" option (SILAC label channels Arg0/Lys0, Arg6/Lys4, Arg10/Lys8) and the "In-Silico Generate Missing Channels" option were enabled. Data were searched against UniProt Homo sapiens (20,365 entries, version from May 2019) in combination with the cRAP database (https://www.thegpm.org/crap/index.htmL) containing common contaminants. For each peptide,

the 3 – 6 most abundant b/y ions were selected for library generation, dependent on their signal intensity. Dynamic retention time alignment was performed based on the high-precision indexed retention time (iRT) concept[79]. Mass tolerances (precursor and fragment ions) as well as peak extraction windows were defined automatically by Spectronaut. Normalisation was disabled and data were filtered at 1% false discovery rate (FDR) on the peptide and protein level ($q$ value < 0.01). High-confidence identifications were exported for further analysis.

## Statistical analysis

Master protein abundance values were analysed by R Studio 4.1.0 (R Core Team, 2020) for data processing, differential expression analysis and visualisation. First, protein abundance values for light (L), medium (M) and heavy (H) channels were log2 transformed and proteins with missing values in any channel were removed. Additionally, proteins of both datasets (mitochondrial proteins and affinity purification via Mge1$_{His}$) with average mean abundances of <3E + 04 and <2.5E + 04 as well as CVs>30% and CV > 50%, respectively, were removed for further analysis. For affinity purifications via mtHsp70$_{His}$, proteins with low signal intensities and high variances (mean abundances of <1E + 04 as well as CV > 50%) were removed. Also, Cox1 was removed due to false positive values for one Cox1 peptide in $\rho^0$ mitochondria. Both datasets were median-normalised (equal median normalisation function, DEqMS 1.12.1) and differentially abundant proteins were determined by a moderated t-test based on linear models (LIMMA 3.50.0[80]). Finally, contrast values were extracted from the linear model for all possible comparisons (H/L, H/M, M/L) and exported to individual result tables. Proteins with a log$_2$-fold change >0.58 and a FDR < 0.05 were defined as significantly regulated.

## Immunoblotting

After polyacrylamide gel electrophoresis, proteins were transferred to polyvinylidene fluoride (PVDF) membranes (EMD Millipore) by semi-dry Western blot in blotting buffer (20% [v/v] ethanol, 20 mM Tris, 150 mM glycine, 0.02% [w/v] SDS). Subsequently, the PVDF membranes were blocked by 5% [w/v] skimmed milk powder in TBST buffer (20 mM Tris/HCl pH 7.4, 12.5 mM NaCl, 0.1% [v/v] Tween-20) for 1 h at room temperature. The incubation steps with primary and secondary antibodies were performed with TBST buffer containing 5% [w/v] skimmed milk powder. The antisera used in this study are listed in Supplementary Table 4. We used enhanced chemiluminescence[81] to detect immunosignals with the image reader LAS 4000 (FujiFilm) or Amersham Imager 680 (Cytiva). Alternatively, immunodetection was performed using the Odyssey CLx Infra-red Imaging System (Li-Cor) and analysed using Image Studio software (Li-Cor). For detection of immune signals with the Licor system, we used secondary antibodies against rabbit coupled to fluorescent labels (IRDye 800CW). The specificity of the immunosignals was confirmed by their absence in cells or mitochondria from deletion strains. In case of essential genes, the size shift of the band in cells expressing tagged-proteins confirmed the specificity of the immunosignal.

## Reproducibility, statistics and image processing

We have shown representative images of biochemical and cell biological experiments like affinity purifications, import assays, blue native electrophoresis, steady state analysis, blue native gels and growth assays. We have performed a certain number of independent replicates of each presented experiment. We indicate here the minimal number of replicates in brackets for the main figure: Figs. 1a (4), 1b (4), 1c (4), 1d (2), 1e (3), 1f (2), 2a (2), 2b (4), 2c (4), 2d (2), 3a (3), 3b (3), 3c (3), 3d (2), 4a (2), 4b (3), 4c (2), 4d (2), 4e (4), 5a (5), 5b (4), 5c (4), 5d (2), 5e (4), 6a (3), 6b (3), 6c (5), 6d (4), 7a (2), 7b (2), 7c (2), 7d (2). Similarly, we performed independent repetitions of the experiments shown in Supplementary Figures. The number of replicates is given in brackets: Supplementary Fig. 1b (2), 1c (2), 1d (3), 1e (2), 1f (2), 2a (2), 2b (2), 3a (3), 3b (3), 4a (2), 4b

(2), 5a (3), 5b (4), 5c (2), 6a (4) and 6b (4). All data figures were processed by Adope Photoshop 2022 and assembled using Graphpad Prism 9.3.1 (471) and Adope Illustrator 2022. Separating white lanes indicate where irrelevant gel lanes were digitally removed. The uncropped files are supplied as Source Data. We provide Supplementary Data 1–3 showing the results of the four replicates of the mass spectrometric data. Signal intensities of protein bands of Western blots or autoradiography were quantified using Multi Gauge (Fujifilm V3.2) and Fuji.Java 8 (Wayne Rasband, National Institute of Health, USA). Single data points of the experiments are provided in the Source Data file.

## Reporting summary

Further information on research design is available in the Nature Portfolio Reporting Summary linked to this article.

## Data availability

The mass spectrometry data have been deposited to the ProteomeXchange Consortium via the PRIDE partner repository and are publicly available under the identifier PXD033024. The authors declare that all data supporting the findings of this study are available within the article or the Supplementary Information. All other data can be obtained by the corresponding authors upon reasonable request. Source data are provided within this paper. Source data are provided with this paper.

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

## Acknowledgements

We thank Drs. N. Pfanner and N. Wiedemann for comments on the manuscript and support. We thank Drs. F. den Brave and W. Voos for critical reading the manuscript. We thank Dr. Robert Hardt for bioinformatic support. This work was supported by grants of the Deutsche Forschungsgemeinschaft (DFG, BE4679/2-2 project ID 269424439 (to T.B.), SFB1218 project ID 269925409 (to T.B.) and FOR2625 WI:4041/3-1 (to D.W.)) and Studienstiftung des Deutschen Volkes (J.S.). The mass spectrometer was funded by the Deutsche Forschungsgemeinschaft (DFG, project number: 386936527).

## Author contributions

Ji.S., L.S., I.S., Ja.S., A.S. and L.B. performed the experiments and analysed the data together with T.B.and D.W.; T.B. designed and supervised the project; T.B. and Ji.S. prepared the figures and wrote the manuscript; all authors discussed results from the experiments and commented on the manuscript.

## Funding

## Competing interests

The authors declare no competing interests.
