## [Peer Review File · Nature Communications]

The mitochondrial Hsp70 controls the assembly of the F1FO-ATP synthaseREVIEWER COMMENTS

Reviewer #1 (Remarks to the Author):

In this manuscript, Song et al. investigate a role of mitochondrial Hsp70 in the biogenesis of the mitochondrial ATP synthase. The electron transport chain generates an electrochemical gradient across the mitochondrial inner membrane, and this proton motive force drives the ATP synthase to synthesize ATP. Although, some aspects of ATP synthase assembly are known, how the F1 catalytic domain is fed into the assembly line remains unclear. The mitochondrial Hsp70 plays a key for import of nuclear-encoded proteins and subsequent folding.

In this study, Song and colleagues address a role of mtHSP70 in the biogenesis of the ATP synthase. The manuscript provides convincing evidence that mtHsp70 couples the binding of newly imported subunits of the F1 catalytic domain (Atp1 and Atp2) and the peripheral stalk subunit Atp5 to their respective assembly factors, and further, monitors their integration to form the mature complex.

This study provides new and interesting insights into how mtHsp70 acts as an assembly and quality control factor and advances our understanding of mitochondrial biogenesis. The work is of high technical standard, very systematic and follows a logical sequence of experiments. Overall, this manuscript addresses a long standing and critical questions, it is of general interest to the field of cell and molecular biology.

Specific comments:

Figs. 1b-d: mtHsp70 is a constituent of the presequence translocase-associated motor and interacts with Pam18, Pam16, and Tim44. In this study, the aforementioned proteins were enriched in the Mge1-His pull-down-mass spectrometry dataset (Supplementary table 2). Hence, together with the TIM23 complex subunits, PAM subunits should also be labeled in the volcano plot in Fig. 1b. Similarly, the western blots in Figs. 1c and d should be decorated with antibodies against Tim44, Pam18 and Pam16.

In Fig. 1d it should be indicated in the legend that this is done in the absence of ATP (0.1 mM EDTA).

In Fig. 1c the quantification is not clear – how was that calculated. Given the reduction of Atp1, 2, and 5 in the total, I would expect that the amounts in the eluate need to be standardized to the total.

Figs. 3b-d: The authors state that the assembly/levels of the F1 domain is reduced in ssc1-62 supporting the idea that mtHsp70 promotes the formation of F1 domain. However, there is a marked increase in

the levels of monomeric ATP synthase in the case of blots for anti-Atp2 and anti-Atp3, and activity staining assay suggesting that this mutant assembles F1 domain to form the monomeric ATP synthase faster than the wild type. How can this be interpreted in the light of the role of mtHSP70?

Figs. 4b and c: The interaction of mtHsp70 with Atp11 or Atp12 is strongly reduced in *ssc1-62* compared to the wild type as observed in Figs. 1b and 4a. Similarly, in Fig. 4b the co-purification of Atp1 and Atp2 is decreased in Atp12 and Atp11 pull-down assays using the *ssc1-62* strain. Based on these results, the authors suggest that mtHsp70 promotes the binding of Atp1 and Atp2 with Atp12 and Atp11 respectively, and these steps are ATP-sensitive. However, this analysis is incomplete without an immuno-decoration using anti-mtHsp70 antibody to verify the levels of eluted mtHsp70 in Figs. 4b and c.

Figs. 5b and c: Similar to the issue discussed above, the western blots in Figs. 5b and c should be decorated with anti-mtHsp70 antibody to support that mtHsp70 promotes the binding of Atp5 with INAC.

Minor points.

TIM is an abbreviation for “translocase of the inner membrane” – “TIM23 translocase” is therefore a pleonasm that should be avoided.

Reviewer #2 (Remarks to the Author):

ATP synthase subunits including F1 alpha and beta as well as the central stalk subunit gamma bind to mHsp70 in both wildtype and rho zero cells. Using a temperature sensitive allele of mHsp70, which does not affect mitochondrial protein import, the authors show that mHsp70 promotes the assembly of the F1 domain of ATP synthase. mHsp70 also interacts with the previously characterized F1 domain assembly factors Atp11 and Atp12. Moreover, pulldowns of Atp11 and Atp12 recover F1 beta and F1 alpha, respectively, in wildtype but not in the background of the temperature sensitive mHsp70 allele. Binding of Atp11 and Atp12 to their corresponding substrates is ATP dependent implying mHsp70 as a mediator. The INAC complex promotes assembly of the peripheral stalk of the ATP synthase. The authors show that deletion of its components results in overaccumulation of Atp5 at mHsp70 suggesting that the latter may handover Atp5 to INAC.

Finally, a mutated version of Atp5 that blocks complete assembly of ATP synthase when expressed in wildtype cells can still be integrated into ATP synthase monomer and dimers in mutant cells carrying a temperature sensitive version of mHsp70 that cannot bind to Atp5. This suggests mHsp70 also functions

in quality control preventing assembly of mutant versions of Atp5 that would interfere with ATP synthase function.

This is an interesting study implicating mHsp70 in two distinct previously not characterized steps of ATP synthase assembly: (i) the formation of the F1 domain together with the assembly factors Atp11 and Atp12 and (ii) the recruitment of Atp5 which links the peripheral stalk with the F1 domain. Finally, if mHsp70 is bypassed, in cell lines expressing a mutant version of the protein, a defective non-functional Atp5 version gets assembled into what will be a non-functional ATP synthase, implying a quality control function for mHsp70.

In general the experiments are carefully designed and the presented results are of high quality and the interpretation of the results are sound.

In my opinion the manuscript is a bit on the short side. While concise writing is a virtue, I think that the non-specialist reader could in some instances, some of which are outlined below, profit from more detailed explanations of the rationales and the results of certain experiment. This can easily be remedied in a revised manuscript.

Major comments

1) Figure S1: Is the F1 domain that accumulates in rho zero cells soluble in the matrix or still somehow associated with the inner membrane?

2) Figure 3 - Import of Atp1 and 2 is assayed in vitro with mitochondria isolated at permissive temperature. However, in the corresponding in vivo experiment Fig. 3c, where cells were incubated at non-permissive temperature, import of the two proteins has not been tested.

3) This manuscript focuses on the function of mHsp70 in ATP synthase assembly. I therefore think it could be useful to provide a non-biased global overview of which proteins bind to wildtype mHsp70 and to compare these results to the temperature sensitive version of mHsp70. Figure 1b provides this type of data for wild-type mHsp70, albeit somewhat indirectly by pull down of Mge1. Is there reason that no direct pulldowns of mHsp70 were analyzed by proteomics?

4) The suggested quality control function of mHsp70 is very interesting. It is stated in the discussion, that mHsp70 could deliver non-productive variants of Atp5 variants for degradation. It would add much to the quality control part of the study, if it would be possible to provide some experimental evidence for this.

5) If think it would help the reader of the involvement of mHsp70 in ATP synthase assembly are graphically described in a model at the end of the manuscript.

Minor comments

- Volcano blots in Fig. 1ab are lacking a complete explanation of the color code. What are the small blue and orange dots?
- Was Atp12 detected in the SILAC pulldown shown in Fig. 1b
- Figure S1: the Tom40 panel on the bottom right is not referred to in the text. The same is the case for the two panels (Tom40, Hsp60) in Figure S2.
- Figure 3a and top page 7 - It is stated that "import of Atp1 or Atp2 was not affected". I don't think this is a precise description. Import is accelerated in ssc1-62 in both cases, especially for Atp2. This should be acknowledged.
- Whenever results with ssc1-62 are shown, please indicated in the legend or the Figure itself whether they were done at permissive or non-permissive temperature (example Fig. S1).
- Page 8: explain better the rationale of using delta fmc1 cells for the non-specialized reader
- INAC should be introduced in some more detail, either in the introduction or the results: How big is it? How does it interact with the membrane? What are the components of INAC?
- Page 8: "INAC promotes the formation of the peripheral stalk, but is not essential for the assembly of the ATP synthase,....". It is not clear to me what is meant by ATP synthase: FoF1 together with the peripheral stalk, or everything except the peripheral stalk, or is meant that INAC promotes the formation of the peripheral stalk, but without it, the stalk can still be formed?
- Fig 5E, lane 3 and 11: please discuss the band between Vm and F1. The same bands are also seen in Fig. S4b.
- Page 9, bottom part: "assembly ofAtp4 and Atp14 was only mildly stimulated in the mutant mitochondria" better "assembly of Atp4 was not and that of Atp14 only mildly stimulated.."

Reviewer #3 (Remarks to the Author):

The paper by Song et al reports on a new role for mtHSP70 as assembly factor in the assembly of the F1FoATPase in the mitochondrial inner membrane. The authors suggest that mtHSP70 works in two levels to fulfil this function (i) it partners with the known assembly factors Atp11 and Atp12 to form the F1 domain of the ATP synthase and (ii) it transfers Atp5 into an orchestrated assembly pathway linking the catalytic head with the peripheral stalk. They also report that mtHsp70 has a role in quality control of the biogenesis of the F1Fo-ATP synthase. These results uncover a new role for mtHSP70, beyond its well established functions in protein import and folding, and the study is overall executed to a high technical standard and presented in a clear manner. I have some minor concerns that I think the authors should address.

1. In the initial experiments showing an association of the ATP subunits to mtHsp70 the authors used a Mge1-His approach, which seems not the most direct one if they were to look for interactors of mtHsp70, compared to using a direct mtHsp70-His binding assay. Why was this? One would have to assume that the interaction of Mge1-mtHsp70 (which is important for the ATPase import motor activity of mtHSP70) is unaffected in rho-zero cells, but is this really the case? They do provide some control experiments with reverse pull downs using mtHsp70-His, but I would think they should have a more direct approach based on mtHsp70 pull-downs.

2. Related to the above point: Do the authors think there are (at least) two separate pools of mtHsp70, one working near the exit of the TIM23/Tim44 import channel as part of the translocation motor and another one near the F1FoATPase? Of note here that the F1FoATPase is thought to be primarily present in the cristae whereas the Tim23 import complex is thought to be primarily present in the boundary IM. Since the authors suggest that it is the mtHsp70 that works as an assembly factor for the F1FoATPase (but not Mge1) the above questions need some clarification.

3. The evidence on the potential quality control role of mtHSP70 is largely based on the use of a single Atp5 mutant (G183A). This needs further support from at least 1-2 additional single point other mutants of Atp5.

4. On the point of quality control of mtHSP70: it is known, and I think generally accepted, that the LON protease is a key protein for the quality control of proteins in the mitochondrial matrix. Have the authors tested whether the potential function of mtHSP70 in a quality control mechanism for the F1FoATPase is independent of LON?

5. It would be worth speculating what happens to the unassembled Atp1/2. If their aggregation cannot be overcome via the interaction with mtHSP70, what happens to these proteins? Are they degraded inside the organelle or retrograde exported to the cytosol for proteasomal degradation?

6. minor point: some grammatical errors, might be worth having another look at the manuscript, for example 4th line from the end of p 5 'no interaction of the soluble ATP synthase subunits have reported' should be 'no interaction of the soluble ATP synthase subunits has been reported'.

NCOMMS-22-17592 by Song et al.

Point-by-point reply to the reviewer's comments

We thank all reviewers for their positive and constructive comments that helped to improve our manuscript. We have added substantial amounts of new experimental data (Figs. 1b, 1d, 3c, 4a, 4c, 7c, 7d, Supplemental Figs. 1d, 1e, 1f, 3 and 6) and made text changes to better describe the presented findings and rationale of the experimental approaches. Furthermore, we have included a new model (Fig. 7e) to summarize the most important findings of our study.

Reviewer #1:

In this manuscript, Song et al. investigate a role of mitochondrial Hsp70 in the biogenesis of the mitochondrial ATP synthase. The electron transport chain generates an electrochemical gradient across the mitochondrial inner membrane, and this proton motive force drives the ATP synthase to synthesize ATP. Although, some aspects of ATP synthase assembly are known, how the F1 catalytic domain is fed into the assembly line remains unclear. The mitochondrial Hsp70 plays a key for import of nuclear-encoded proteins and subsequent folding.

In this study, Song and colleagues address a role of mtHSP70 in the biogenesis of the ATP synthase. The manuscript provides convincing evidence that mtHsp70 couples the binding of newly imported subunits of the F1 catalytic domain (Atp1 and Atp2) and the peripheral stalk subunit Atp5 to their respective assembly factors, and further, monitors their integration to form the mature complex.

This study provides new and interesting insights into how mtHsp70 acts as an assembly and quality control factor and advances our understanding of mitochondrial biogenesis. The work is of high technical standard, very systematic and follows a logical sequence of experiments. Overall, this manuscript addresses a long standing and critical questions, it is of general interest to the field of cell and molecular biology.

We thank the reviewer for the positive and constructive comments that helped to improve our manuscript.

Figs. 1b-d: mtHsp70 is a constituent of the presequence translocase-associated motor and interacts with Pam18, Pam16, and Tim44. In this study, the aforementioned proteins were enriched in the Mge1-His pull-down-mass spectrometry dataset (Supplementary table 2). Hence, together with the TIM23 complex subunits, PAM subunits should also be labeled in the volcano plot in Fig. 1b. Similarly, the western blots in Figs. 1c and d should be decorated with antibodies against Tim44, Pam18 and Pam16.

We have now labelled TIM23 and PAM subunits in the volcano plots from both the novel pulldown via mtHsp70_{His} and the previously included pulldown using a Mge1_{His}

affinity matrix (novel Fig. 1b and 1c). In addition, we added western blots for Pam16, Pam18 and Tim44 from pull-downs via mtHsp70His and the Mge1His affinity matrix (Fig. 1d and 1e).

In Fig. 1d it should be indicated in the legend that this is done in the absence of ATP (0.1 mM EDTA).

We added a statement to the Figure legend that the pull-down was performed in the absence of ATP.

In Fig. 1c the quantification is not clear – how was that calculated. Given the reduction of Atp1, 2, and 5 in the total, I would expect that the amounts in the eluate need to be standardized to the total.

We first determined the fraction of the total mitochondrial Atp1, Atp2 and Atp5 amount that was co-eluted with Mge1_{His} in wild-type and rho zero mitochondria. In this step, the reduced amounts in the load were taken into account. The determined fraction of Mge1-bound proteins in wild-type mitochondria was set to 100%. Subsequently, the ratio between the fraction of rho0 and wild-type mitochondria was calculated. We included now a more detailed description of the quantification in the legend of figure 1e.

Figs. 3b-d: The authors state that the assembly/levels of the F₁ domain is reduced in *ssc1-62* supporting the idea that mtHsp70 promotes the formation of F₁ domain. However, there is a marked increase in the levels of monomeric ATP synthase in the case of blots for anti-Atp2 and anti-Atp3, and activity staining assay suggesting that this mutant assembles F₁ domain to form the monomeric ATP synthase faster than the wild type. How can this be interpreted in the light of the role of mtHSP70?

We quantified the amounts of monomeric and dimeric ATP synthase and the assembled F₁ domain in *ssc1-62* mitochondria that were isolated from heat stressed cells (novel Fig. 3c). The quantification revealed a strong reduction of the F₁-domain, while monomer and dimer were moderately affected. This observation supports our conclusion that mtHsp70 promotes the formation of the F₁ domain. We observed a moderate increase in the monomer and a moderate decrease of the dimer of the ATP synthase, which reflects perturbation of the assembly of the ATP synthase by impaired mtHsp70 function and points to an important function of mtHsp70 for the formation of the ATP synthase.

Figs. 4b and c: The interaction of mtHsp70 with Atp11 or Atp12 is strongly reduced in *ssc1-62* compared to the wild type as observed in Figs. 1b and 4a. Similarly, in Fig. 4b the co-purification of Atp1 and Atp2 is decreased in Atp12 and Atp11 pull-down assays using the *ssc1-62* strain. Based on these results, the authors suggest that mtHsp70 promotes the binding of Atp1 and Atp2 with Atp12 and Atp11 respectively, and these steps are ATP-

sensitive. However, this analysis is incomplete without an immuno-decoration using anti-mtHsp70 antibody to verify the levels of eluted mtHsp70 in Figs. 4b and c.

We provide in the revised manuscript several lines of evidence that Atp12 and Atp11 interact with mtHsp70. First, Atp11 and Atp12 were co-purified with increased efficiency via mtHsp70_{His} and Mge1_{His} in rho zero mitochondria as shown by mass spectrometry (Fig. 1b and 1c). Second, we could show via Western blotting that Atp12 binds to mtHsp70 (Fig. 1d). Third, we found that co-purification of Atp12 via Mge1_{His} is diminished in *ssc1-62* mitochondria (Fig. 4a). Finally, we show in the new Fig. 4c that the co-purification of mtHsp70 with His-tagged Atp11 and His-tagged Atp12 was blocked upon preincubation with ATP (Fig. 4c). mtHsp70 is present in 100.000 - 130.000 copies in the cell, while Atp11 and Atp12 are present in 1.100-1.800 copies per cell (Morgenstern et al., 2017). Therefore, only a low percentage of mtHsp70 was found in the elution fractions of His-tagged Atp11 and Atp12.

Figs. 5b and c: Similar to the issue discussed above, the western blots in Figs. 5b and c should be decorated with anti-mtHsp70 antibody to support that mtHsp70 promotes the binding of Atp5 with INAC.

According to our model, mtHsp70 delivers Atp5 to the assembly intermediate at INAC to allow the linkage of the peripheral stalk to the F₁ domain. Furthermore, INAC subunits are present in 100-250 copies per cell, while mtHsp70 is present in 100.000 – 120.000 copies per cell (Morgenstern et al., 2017). Therefore, we likely could not detect mtHsp70 in the elution samples of pulldowns via His-tagged Ina17 and Ina22 (data not shown). The experimental data shown in Fig. 4b and 4c indicate that mtHsp70 stabilizes the binding of Atp5 and consequently Atp1/Atp2 to INAC. In contrast, the membrane-anchored Atp4 binds to INAC, which is not affected upon preincubation with ATP. We adjusted the text to make this point clear (Page 11, first paragraph).

TIM is an abbreviation for “translocase of the inner membrane” – “TIM23 translocase” is therefore a pleonasm that should be avoided.

We corrected “TIM23 translocase” to “TIM23 complex”.

Reviewer #2:

ATP synthase subunits including F1 alpha and beta as well as the central stalk subunit gamma bind to mHsp70 in both wildtype and rho zero cells. Using a temperature sensitive allele of mHsp70, which does not affect mitochondrial protein import, the authors show that mHsp70 promotes the assembly of the F1 domain of ATP synthase. mHsp70 also interacts with the previously characterized F1 domain assembly factors Atp11 and Atp12. Moreover, pulldowns of Atp11 and Atp12 recover F1 beta and F1 alpha, respectively, in wildtype but not in the background of the temperature sensitive mHsp70 allele. Binding of Atp11 and Atp12 to their

corresponding substrates is ATP dependent implying mHsp70 as a mediator. The INAC complex promotes assembly of the peripheral stalk of the ATP synthase. The authors show that deletion of its components results in overaccumulation of Atp5 at mHsp70 suggesting that the latter may handover Atp5 to INAC.

Finally, a mutated version of Atp5 that blocks complete assembly of ATP synthase when expressed in wildtype cells can still be integrated into ATP synthase monomer and dimers in mutant cells carrying a temperature sensitive version of mHsp70 that cannot bind to Atp5. This suggests mHsp70 also functions in quality control preventing assembly of mutant versions of Atp5 that would interfere with ATP synthase function.

This is an interesting study implicating mHsp70 in two distinct previously not characterized steps of ATP synthase assembly: (i) the formation of the F1 domain together with the assembly factors Atp11 and Atp12 and (ii) the recruitment of Atp5 which links the peripheral stalk with the F1 domain. Finally, if mHsp70 is bypassed, in cell lines expressing a mutant version of the protein, a defective non-functional Atp5 version gets assembled into what will be a non-functional ATP synthase, implying a quality control function for mHsp70. In general the experiments are carefully designed and the presented results are of high quality and the interpretation of the results are sound.

In my opinion the manuscript is a bit on the short side. While concise writing is a virtue, I think that the non-specialist reader could in some instances, some of which are outlined below, profit from more detailed explanations of the rationales and the results of certain experiment. This can easily be remedied in a revised manuscript.

We thank the reviewer for the constructive comments, which helped to improve the manuscript. We have included text changes in the revised manuscript to explain the rationale behind the experimental strategy and the obtained results.

Major comments

1) Figure S1: Is the F1 domain that accumulates in rho zero cells soluble in the matrix or still somehow associated with the inner membrane?

We separated total membrane and soluble fractions of osmotically ruptured mitochondria. Our results show that the F₁ domain is present in the soluble fraction in rho zero mitochondria (Novel Supplemental Fig. 1d) and therefore is attached to the inner membrane.

2) Figure 3 - Import of Atp1 and 2 is assayed in vitro with mitochondria isolated at permissive temperature. However, in the corresponding in vivo experiment Fig. 3c, where cells were incubated at non-permissive temperature, import of the two proteins has not been tested.

We have tested the import of Atp1 and Atp2 into *ssc1-62* mitochondria isolated from cells after in vivo heat shock. The import and consequently the assembly of both proteins is impaired (Novel Supplemental Fig. 3). Import defects were not observed when mitochondria were isolated from non-stressed cells (Figs. 3a and 6a). Therefore,

these mitochondria were used to study the assembly of ATP synthase (Figs. 3b and 6b).

3) This manuscript focuses on the function of mtHsp70 in ATP synthase assembly. I therefore think it could be useful to provide a non-biased global overview of which proteins bind to wildtype mtHsp70 and to compare these results to the temperature sensitive version of mtHsp70. Figure 1b provides this type of data for wild-type mtHsp70, albeit somewhat indirectly by pull down of Mge1. Is there reason that no direct pulldowns of mtHsp70 were analyzed by proteomics?

As suggested by the reviewer, we performed an affinity purification via His-tagged mtHsp70 in wild-type and rho zero mitochondria and directly compared the elution fractions (Novel Fig. 1b). We detected a similar enrichment of subunits of the ATP synthase in mitochondria from rho zero cells like we detected in the Mge1_{His} pulldown (Fig. 1c). We also tried to His-tag the Ssc1-62 variant. However, the cells were not viable on respiratory growth conditions. Therefore, we used the Mge1_{His} affinity matrix to analyse whether the binding of partner proteins to mtHsp70 was affected (Novel Fig. 4a). We found that the Ssc1-62 variant binds with a similar efficiency to the Mge1_{His} affinity matrix like the wild-type mtHsp70. However, the co-purification of Atp1, Atp2, Atp5 and Atp12 was diminished. Furthermore, the binding of TIM23 and PAM subunits was decreased (Fig. 4a). Thus, the mutated mtHsp70 (*ssc1-62*) interacts with Mge1, but affects the association of TIM23, PAM and ATP synthase.

4) The suggested quality control function of mtHsp70 is very interesting. It is stated in the discussion, that mtHsp70 could deliver non-productive variants of Atp5 variants for degradation. It would add much to the quality control part of the study, if it would be possible to provide some experimental evidence for this.

In the revised version, we provide evidence that mtHsp70 could deliver unassembled Atp5 to degradation. We found that overexpressed HA-tagged Atp5 and Atp5G183A is stabilized in *ssc1-62* mitochondria (Fig. 7c), where its binding to mtHsp70 is impaired (Fig. 7a). Furthermore, we found that mtHsp70 is stabilized in *pim1Δ*, but no in *yta12Δ* cells. Pim1 was reported previously to degrade unassembled Atp1, Atp2 and Atp7 (Suzuki et al., 1994; Bayot et al., 2010). Our findings indicate that mtHsp70 cooperates with Pim1 in the degradation of Atp5. A similar cooperation of mtHsp70 and Pim1 was reported before for the degradation of misfolded proteins (Savel'ev et al., 1998).

5) If think it would help the reader of the involvement of mtHsp70 in ATP synthase assembly are graphically described in a model at the end of the manuscript.

As suggested by the reviewer, we included in the revised version a hypothetical model about the role of mtHsp70 in the assembly of the ATP synthase (Novel Fig. 7e).

Minor comments

- Volcano blots in Fig. 1ab are lacking a complete explanation of the color code. What are the small blue and orange dots?

We have changed the color code of the dots. All blue and orange dots are now shown in grey to avoid confusion. Previously, the colored dots marked significantly decreased and increased proteins.

- Was ATP12 detected in the SILAC pulldown shown in Fig. 1b

We did not detect Atp12 in this pulldown for unknown reasons. We have included a purification via mtHsp70_{His} in the revised manuscript. Here, we detected Atp12 both via mass spectrometry and via Western blotting (Novel Figs. 1b and 1d). Furthermore, we found Atp12 co-purified via a Mge1_{His} affinity matrix (Fig. 4a). All these data reveal an interaction between mtHsp70 and Atp12.

- Figure S1: the Tom40 panel on the bottom right is not referred to in the text. The same is the case for the two panels (Tom40, Hsp60) in Figure S2.

We now refer to the blots in the revised manuscript.

- Figure 3a and top page 7 - It is stated that "import of Atp1 or Atp2 was not affected". I don't think this is a precise description. Import is accelerated in *ssc1-62* in both cases, especially for Atp2. This should be acknowledged.

We changed the statement to "We found that the import of both ATP synthase subunits into in *ssc1-62* mitochondria was mildly accelerated".

- Whenever results with *ssc1-62* are shown, please indicated in the legend or the Figure itself whether they were done at permissive or non-permissive temperature (example Fig. S1).

We have now indicated in the figure legend whether the *ssc1-62* cells were grown under permissive or non-permissive conditions. Experiments performed in Figs. 3c, 3d, 4b and Supplemental Fig. 3 are the only experiments that were performed with mitochondria isolated from heat-stressed *ssc1-62* cells. All other experiments were performed with mitochondria isolated from *ssc1-62* cells grown under permissive conditions. Levels of the detected mitochondrial proteins and protein complexes were unchanged under these conditions (Supplementary Fig. 2).

- Page 8: explain better the rationale of using delta *fmc1* cells for the non-specialized reader

We used *fmc1*Δ cells here to investigate the cross-talk of mtHsp70 with Atp12. Deletion of Atp12 blocks formation of the ATP synthase and blocks respiratory growth.

Therefore, we used the *fmc1*Δ strain since Fmc1 is required to stabilize Atp12 (Lefebvre-Legendre et al., 2001). Indeed, we found that deleting Fmc1 in the *ssc1-62* mutant blocks growth of the cells and leads to reduced formation of the ATP synthase. Supporting a functional link between mtHsp70 and Atp11/Atp12, we found that His-tagging of Atp11 or Atp12 impairs formation of the ATP synthase and growth on respiratory media. Based on these observations and interaction studies, we conclude that mtHsp70 cooperates with Atp11/Atp12 in the formation of the F₁ domain. We provide now text changes to better explain the usage the ratio behind using the *fmc1*Δ strain in this study (Page 10, first paragraph).

- INAC should be introduced in some more detail, either in the introduction or the results: How big is it? How does it interact with the membrane? What are the components of INAC?

We added the requested information in the introduction: “The inner membrane assembly complex (INAC) promotes the formation of the peripheral stalk and facilitates its association with the membrane-bound rotor module (Lytovchenko et al., 2013; Naumenko et al., 2017). INAC consists of the two single-spanning inner membrane proteins Ina17 and Ina22 that form a protein complex of around 300 kDa on blue native gels (Lytovchenko et al., 2013; Naumenko et al., 2017)”

- Page 8: “INAC promotes the formation of the peripheral stalk, but is not essential for the assembly of the ATP synthase,....”. It is not clear to me what is meant by ATP synthase: FoF1 together with the peripheral stalk, or everything except the peripheral stalk, or is meant that INAC promotes the formation of the peripheral stalk, but without it, the stalk can still be formed?

Lytovchenko and colleagues reported that INAC assembles the peripheral stalk that links the F₁ domain to the membrane embedded FO domain (Lytovchenko et al., 2013 EMBO J.). However, the formation of the mature ATP synthase is not fully blocked in the absence of a functional INAC (Fig. 5e), indicating that further mechanisms exist. We report here that deleting *Ina17* in *ssc1-62* blocks respiratory growth and formation of the ATP synthase (Fig. 5d and 5e), pointing to an important role of the chaperone in this critical assembly steps of the ATP synthase.

- Fig 5E, lane 3 and 11: please discuss the band between Vm and F1. The same bands are also seen in Fig. S4b.

We detected an unknown complex band that contains Atp1 and Atp2 and run between the monomer and the F₁ domain in mitochondria from *ina17*Δ cells (Fig. 5e and Supplemental Fig. 5b). We now marked this band in the figures and mentioned it in the text. The identity of this band remains unknown. It will be interesting to define its composition and function in future studies.

- Page 9, bottom part: “assembly ofAtp4 and Atp14 was only mildly stimulated in the

mutant mitochondria” better “assembly of Atp4 was not and that of Atp14 only mildly stimulated..”

We changed the text following the reviewer’s suggestion to: “For comparison, the assembly of Atp4 was not affected and the assembly of Atp14 was mildly stimulated in the mutant mitochondria (Figs. 6a and 6b).”

Reviewer #3:

The paper by Song et al reports on a new role for mtHSP70 as assembly factor in the assembly of the F1FoATPase in the mitochondrial inner membrane. The authors suggest that mtHSP70 works in two levels to fulfil this function (i) it partners with the known assembly factors Atp11 and Atp12 to form the F1 domain of the ATP synthase and (ii) it transfers Atp5 into an orchestrated assembly pathway linking the catalytic head with the peripheral stalk. They also report that mtHsp70 has a role in quality control of the biogenesis of the F1Fo-ATP synthase. These results uncover a new role for mtHSP70, beyond its well established functions in protein import and folding, and the study is overall executed to a high technical standard and presented in a clear manner. I have some minor concerns that I think the authors should address.

We thank the reviewer for the positive and constructive comments that helped to improve our manuscript.

1. In the initial experiments showing an association of the ATP subunits to mtHsp70 the authors used a Mge1-His approach, which seems not the most direct one if they were to look for interactors of mtHsp70, compared to using a direct mtHsp70-His binding assay. Why was this? One would have to assume that the interaction of Mge1-mtHsp70 (which is important for the ATPase import motor activity of mtHSP70) is unaffected in rho-zero cells, but is this really the case? They do provide some control experiments with reverse pull downs using mtHsp70-His, but I would think they should have a more direct approach based on mtHsp70 pull-downs.

Following the suggestions of reviewers 2 and 3, we performed an affinity purification via His-tagged mtHsp70 in mitochondria from rho zero cells and analysed it by SILAC-based mass spectrometry (Novel Fig. 1b). We compare the co-purification of mitochondrial proteins in rho zero mitochondria with wild-type mitochondria. Similar to the pulldown via Mge1_{His}, we found an enrichment of several ATP synthase subunits in the eluate (Novel Fig. 1b and 1c). We could confirm these data by Western blotting (Novel Fig. 1d). The binding of Mge1 to mtHsp70 is not affected in mitochondria from rho zero cells.

2. Related to the above point: Do they authors think there are (at least) two separate pools of mtHsp70, one working near the exit of the TIM23/Tim44 import channel as part of the translocation motor and another one near the F1FoATPase? Of note here that the F1FoATPase is thought to be primarily present in the cristae whereas the Tim23 import complex is thought to be primarily present in the boundary IM. Since the authors suggest that it is the mtHsp70

that works as an assembly factor for the F1FoATPase (but not Mge1) the above questions need some clarification.

We tested the possibility of separate mtHsp70 pools by performing pulldown via His-tagged Tim17 and His-tagged Atp1 (Novel Supplemental Figs. 1e and 1f). We could not co-purify Atp1, Atp2 or Atp5 with His-tagged Tim17. Conversely, we could not co-purify the subunit Tim17, Tim23 and Tim44 of the TIM23 complex with His-tagged Atp1 (Novel Supplemental Figs. 1e and 1f). Based on these observations, we conclude that the pools mtHsp70 that perform protein import and assembly of the ATP synthase are distinct.

3. The evidence on the potential quality control role of mtHSP70 is largely based on the use of a single Atp5 mutant (G183A). This needs further support from at least 1-2 additional single point other mutants of Atp5.

We searched for amino acid residues that are located in the interface of Atp5 and the F1 domain. Reinders et al reported two phosphosites in Atp5 (S48 and T139) (Reinders et al., 2007). The available structure of the yeast ATP synthase shows that T139 localizes close to Atp1 of the F₁-domain (Srivastava et al., 2018). We mutated both residues to either glutamate or alanine (Atp5^{S48A T139A} and Atp5^{S48E T139E}). The assembly of both Atp5 variants was impaired in comparison to the assembly of the wild-type Atp5 (Novel Supplemental Fig. 6a). However, the assembly of both variants occurred more efficiently in the *ssc1-62* mutant mitochondria (Novel Supplemental Fig. 6b). Thus, three assembly-defective Atp5 variants integrated with higher efficiency into the ATP synthase in *ssc1-62*, pointing to a quality control of mtHsp70. In addition, we show that HA-tagged Atp5 and Atp5^{G183A} accumulate in *ssc1-62* and *pim1Δ* cells, indicating that their removal is impaired (see also comments to reviewer 2). Supporting these findings, mtHsp70 was reported previously to cooperate with Pim1 to degrade misfolded proteins (Savel'el et al., 1998). Altogether, these observations reveal a function of mtHsp70 in quality control of Atp5.

4. On the point of quality control of mtHSP70: it is known, and I think generally accepted, that the LON protease is a key protein for the quality control of proteins in the mitochondrial matrix. Have the authors tested whether the potential function of mtHSP70 in a quality control mechanism for the F1FoATPase is independent of LON?

We have analysed how unassembled Atp5 is degraded (compare also response to reviewer 2). We found that expressed HA-tagged Atp5 and Atp5^{G183A} are stabilized in *ssc1-62* and *pim1Δ* cells (Figs. 7c and 7d). As control, loss of Yta12 that is part of the m-AAA did not affect the degradation of the Atp5 variants. We conclude that mtHsp70 cooperates with Pim1/LON in the degradation of Atp5.

5. It would be worth speculating what happens to the unassembled Atp1/2. If their aggregation cannot be overcome via the interaction with mtHSP70, what happens to

these proteins? Are they degraded inside the organelle or retrograde exported to the cytosol for proteasomal degradation?

It was reported previously that Atp1 and Atp2 are substrates of Pim1/LON (Suzuki et al., 1994; Bayot et al., 2010). MtHsp70 cooperates with Pim1/LON to degrade misfolded proteins (Savel'ev et al., 1998). Therefore, it is likely that Atp1 and Atp2 are degraded inside mitochondria. We discuss the role of Pim1 for the degradation of subunits of the ATP synthase and its cooperation with mtHsp70 in the discussion of the revised manuscript (PAGE 14, second paragraph).

6. minor point: some grammatical errors, might be worth having another look at the manuscript, for example 4th line from the end of p 5 'no interaction of the soluble ATP synthase subunits have reported' should be 'no interaction of the soluble ATP synthase subunits has been reported'.

We checked carefully the entire text for wording and grammatical errors.

REVIEWERS' COMMENTS

Reviewer #1 (Remarks to the Author):

The authors have satisfyingly addressed all points raised on the previous version. They have performed additional experiments, and have improved the manuscript. Further, the authors have added a new figure (Fig. 7e) to illustrate a working model for the role of mtHsp70 in the biogenesis of mitochondrial ATP synthase which makes it clearer to the reader.

As mentioned previously, this work showcases the multifaceted functions of Hsp70 in mitochondrial protein biogenesis and it is of general interest to the fields of cell and molecular biology.

The manuscript should be accepted for publication with the following changes:

a. In page 6, "... We did not observe any binding of Tim23 subunits to Atp1His (Supplementary Figs. 1e and 1f)." should be changed to "...TIM23 complex subunits...".

b. In Fig. 3d, the label "ATP synthase activity stain" should be corrected – y and n overlapped.

Reviewer #2 (Remarks to the Author):

The authors have adequately all questions, comments and criticisms of my review. - This is an exciting manuscript that fills an important gap in the ATP synthase assembly pathway. I especially liked Fig. 7e of the revised manuscript: the authors did an excellent job presenting their their model for ATP synthase assembly.

Reviewer #3 (Remarks to the Author):

The authors have provided a substantially improved manuscript with much additional data and new figures addressing satisfactorily the points I raised in my original review. I am happy to recommend acceptance of this revised version.

NCOMMS-22-17592 by Song et al.

Point-by-point reply to the reviewer's comments:

Reviewer 1:

The authors have satisfyingly addressed all points raised on the previous version. They have performed additional experiments, and have improved the manuscript. Further, the authors have added a new figure (Fig. 7e) to illustrate a working model for the role of mtHsp70 in the biogenesis of mitochondrial ATP synthase which makes it clearer to the reader.

As mentioned previously, this work showcases the multifaceted functions of Hsp70 in mitochondrial protein biogenesis and it is of general interest to the fields of cell and molecular biology.

The manuscript should be accepted for publication with the following changes:

a. In page 6, "... We did not observe any binding of Tim23 subunits to Atp1His (Supplementary Figs. 1e and 1f)." should be changed to "...TIM23 complex subunits...".

We thank the reviewer for her/his positive response. We have corrected the statement to "...TIM23 complex subunits..." as requested (Page 6).

b. In Fig. 3d, the label "ATP synthase activity stain" should be corrected – y and n overlapped.

The mistake in the labelling occurred during formatting of the Illustrator file into a PDF file. We hope that this time it will not occur since we upload the original ai files of the figures.

Reviewer 2:

The authors have adequately all questions, comments and criticisms of my review. - This is an exciting manuscript that fills an important gap in the ATP synthase assembly pathway. I especially liked Fig. 7e of the revised manuscript: the authors did an excellent job presenting their model for ATP synthase assembly.

We thank the reviewer for her/his positive response.

Reviewer 3:

The authors have provided a substantially improved manuscript with much additional data and new figures addressing satisfactorily the points I raised in my original review. I am happy to recommend acceptance of this revised version.

We thank the reviewer for her/his positive response.